# Cell volume regulates terminal differentiation of cultured human epidermal keratinocytes

Sebastiaan Zijl[1,*], Toru Hiratsuka[1,2,3], Atefeh Mobasseri[1], Mirsana Ebrahimkutty[4], Mandy Börmel[5], Sergi Garcia-Manyes[6,7] and Fiona M. Watt[1,4,‡]

## ABSTRACT

To gain insights into the human epidermal stem cell niche, we have previously identified micron-scale topographical substrates that regulate differentiation of spread keratinocytes. On one substrate (S1), cells interact with circular topographies and differentiation is stimulated; on the other (S2), cells interact with triangular topographies and differentiation is inhibited. Cell stiffness on S1 and S2 was similar, and nuclear localisation of the mechano-sensitive transcriptional regulator YAP1 was decreased on S1 and S2 compared to on flat substrates. However, cells on S2 exhibited reduced cell volume, leading us to explore the potential for volume-regulated differentiation. Treatment with polyethylene glycol decreased cell volume and inhibited differentiation under a range of conditions. Conversely, deionized water increased cell volume and stimulated differentiation. Bulk RNA sequencing identified several substrate-responsive genes, including aquaporins and ion channels. A membrane permeable $Ca^{2+}$ chelator and an inhibitor of the water channel aquaporin 3 blocked volume-induced differentiation. These studies identify cell volume as a mechanism by which keratinocyte–niche interactions regulate terminal differentiation.

KEY WORDS: Stem cell, Niche, Volume control, Cell adhesion, Differentiation, Epidermis

## INTRODUCTION

Human skin comprises the epidermis, which is formed of multiple layers of epithelial cells called keratinocytes, and the underlying connective tissue, the dermis (Rognoni and Watt, 2018). The epidermis and dermis are separated by a basement membrane, comprised of extracellular matrix proteins such as collagen-IV (Watt and Fujiwara, 2011). The basal epidermal layer, which is attached to the basement membrane, contains a patterned distribution of stem cells and cells that have initiated terminal differentiation. Differentiating cells detach from the basement membrane and move through the suprabasal epidermal layers towards the skin surface, from which they are shed (Zijl et al., 2022).

Human epidermal keratinocytes can be grown in culture under conditions that support the maintenance of stem cells and the process of terminal differentiation (Zijl et al., 2022). Cultured human epidermis can therefore be used to examine, at single-cell resolution, how individual stem cells make fate decisions based on external cues from the local microenvironment (Watt, 2016; Louis et al., 2022; Negri and Watt, 2022; Negri et al., 2023).

Previous studies have highlighted the role of cell–substrate interactions in controlling the differentiation of human epidermal stem cells (Zijl et al., 2022). When single cells are seeded on extracellular matrix (ECM)-coated micro-patterned islands, for example, differentiation is triggered by restricted spreading. This process depends on the ratio of F- to G-actin and activation of serum response factor (SRF) and its co-factor megakaryocytic acute leukaemia (MAL) (Watt et al., 1988; Connelly et al., 2010). Differentiation is also triggered when cells are plated on ECM-coated soft hydrogels or on hydrogel–nanoparticle composites with high nanoparticle spacing (Trappmann et al., 2012). Mechanistically, this is mediated via the downregulation of the extracellular signal-regulated kinase (ERK)/mitogen-activated protein kinase (MAPK) pathway.

Although keratinocyte differentiation is typically associated with reduced cell spreading, we have found that micron-scale substrate topographies can also promote the differentiation of spread cells (Zijl et al., 2019). A substrate comprising circular topographical features of ∼3 μm diameter, 5 μm height and unequal spacing (designated S1) promotes differentiation of spread cells via a mechanism that is blocked by Rho kinase inhibition or treatment with the myosin II inhibitor blebbistatin, but not by SRF inhibition. Conversely, a substrate comprising regularly spaced right-angled triangles (8 μm sides) of 5 μm height (designated S2) suppresses differentiation relative to flat surfaces (Zijl et al., 2019). In the present study, we have further explored how keratinocytes respond to each of these substrates, uncovering a potential role for cell volume in the regulation of keratinocyte differentiation.

## RESULTS

### Keratinocytes on S1 substrates can initiate differentiation while spread

To enrich for undifferentiated keratinocytes (epidermal stem cells) we seeded single-cell suspensions of keratinocytes, comprising a mixture of basal and differentiated cells, on collagen-coated S1, S2 or flat surfaces for 1 h and washed off the non-adherent cells (Jones

[1]Centre for Gene Therapy and Regenerative Medicine, King's College London, 28th Floor, Tower Wing, Guy's Hospital, Great Maze Pond, London SE1 9RT, UK. [2]Department of Molecular Oncology, Graduate School of Medicine, Osaka University, Osaka 541-8567, Japan. [3]Department of Oncogenesis and Growth Regulation, Research Center, Osaka International Cancer Institute, Osaka 541-8567, Japan. [4]Directors' Unit, EMBL, Meyerhofstr. 1, 69117 Heidelberg, Germany. [5]EMBL Electron Microscopy Core Facility, Meyerhofstr. 1, 69117 Heidelberg, Germany. [6]Department of Physics, Randall Centre for Cell and Molecular Biophysics, Centre for the Physical Science of Life and London Centre for Nanotechnology, King's College London, London SE1 9RT, UK. [7]Single Molecule Mechanobiology Laboratory, The Francis Crick Institute, London NW1 1AT, UK.
*Present address: Gurdon Institute, University of Cambridge, Tennis Court Road, Cambridge CB2 1QN, UK.

‡Author for correspondence (fiona.watt@embo.org)

S.Z., 0000-0002-2856-1331; T.H., 0000-0002-5359-2690; M.E., 0009-0005-8738-7547; M.B., 0000-0002-4354-891X; S.G.-M, 0000-0001-5140-2606; F.M.W., 0000-0001-9151-5154

and Watt, 1993; Zijl et al., 2019). Scanning electron microscope (SEM) images of keratinocytes 24 h after plating (Fig. 1A–F) revealed that cells on S1 substrates spread over the individual pillars (Fig. 1C,D). Some pillars at the edges of the substrates were bent, suggesting that they had been subject to pulling forces by the cells (Fig. 1D; Eyckmans et al., 2011), although artefacts caused by sample preparation cannot be ruled out entirely. The cytoplasm of cells on S2 substrates appeared to be draped over the triangular features (Fig. 1E,F). In some cells on S2 substrates, the nucleus extended above the surface of the features (Fig. 1F), whereas in other cells the nucleus appeared to have been accommodated in-between the features (Fig. 1E), consistent with nuclear distortions observed by light microscopy (Zijl et al., 2019). Heterogeneity in terms of flattened versus protruding nuclei was also evident on flat and S1 substrates (Fig. 1A–D).

To confirm that cells seeded on S1 initiated differentiation without rounding (Connelly et al., 2010), we performed live-cell imaging of keratinocytes transduced with a pLenti-IVL-mCherry-LifeAct-EGFP reporter. This reporter expresses LifeAct–EGFP, a peptide that binds to actin filaments (Riedl et al., 2008; Belin et al., 2014), under the control of the human PGK promoter, and expresses mCherry under the control of the involucrin (IVL) promoter (Hiratsuka et al., 2020) (Fig. 2A). Flow cytometry confirmed that mCherry was expressed by cells of high forward and side scatter (enriched for differentiating cells; Jones and Watt, 1993; Connelly et al., 2010) and that mCherry-positive cells co-expressed LifeAct–EGFP (Fig. 2A). We flow sorted LifeActGFP+ cells (Fig. 2A) and seeded them on the S1 substrates (Fig. 2B).

Cells expressing the IVL-mCherry-LifeAct-EGFP plasmid were imaged at 1 h intervals starting 4–6 h after seeding. Fig. 2B shows representative images of a single field of keratinocytes obtained from live-cell imaging experiments. At 6 h, several cells in the field were already expressing mCherry and exhibited variable morphologies. Two cells (arrowheads) were mCherry-negative at 6 h and 9 h but expressed mCherry at 12 h and 15 h; in each case the onset of mCherry expression occurred while the cells were spread.

Using LifeAct–EGFP to visualize individual cells, whether or not they had differentiated, we measured the mean mCherry fluorescence of 30 cells on S1 that had upregulated mCherry by 20 h (Fig. 2C). The mean mCherry signal from cells increased above baseline from 12 h onwards (Fig. 2D), consistent with earlier studies on the kinetics of keratinocyte differentiation (Connelly et al., 2010; Zijl et al., 2019; Hiratsuka et al., 2020). These observations establish that cells plated on S1 can upregulate involucrin expression while being spread.

### RNAseq does not reveal unique features of differentiation on S1

Changes in gene expression associated with commitment to differentiation have previously been described in keratinocytes undergoing differentiation in single-cell suspension (Mishra et al., 2017), including a network of interacting protein phosphatases that are also upregulated during the basal to suprabasal transition of keratinocytes *in vivo* (Reynolds et al., 2021; Negri et al., 2023). To examine whether there was a unique gene expression programme associated with differentiation on S1, we seeded keratinocytes on S1 or S2 substrates and extracted RNA at 1 h (initial adhesion), 4 h (differentiation commitment; Mishra et al., 2017) and 12 h (IVL expression on S1; Fig. 2D). RNA was harvested from three

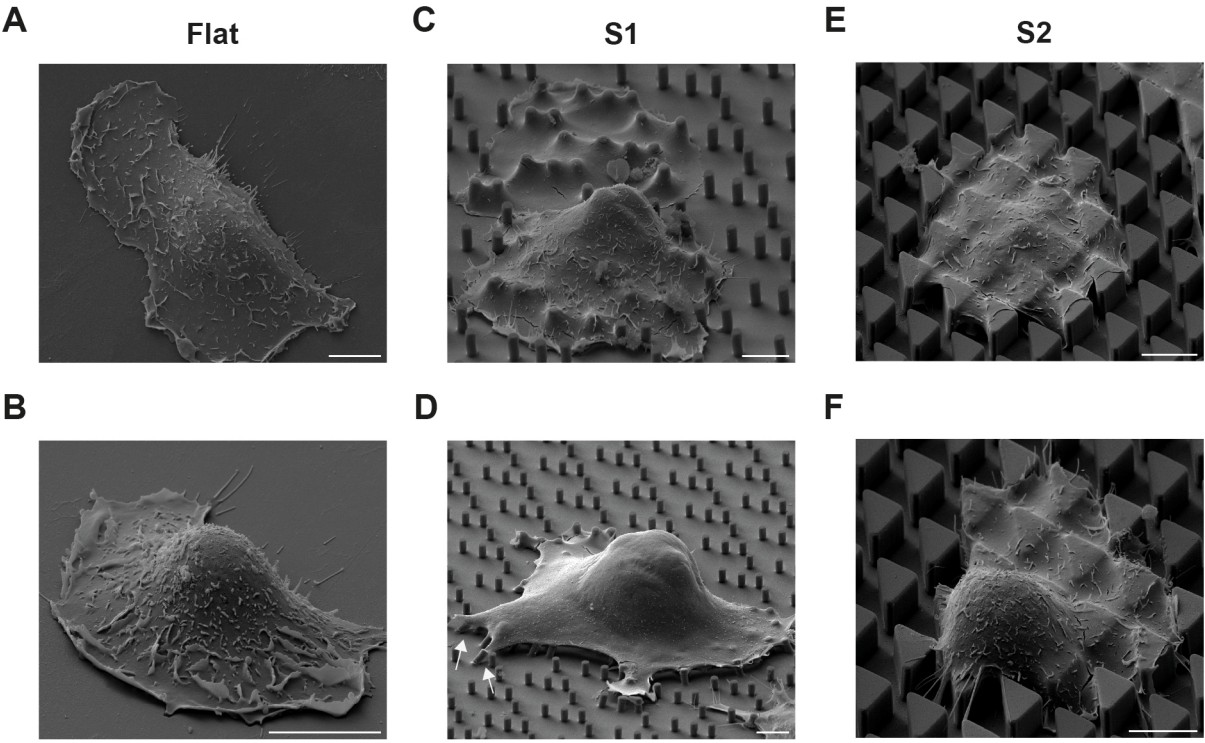

**Fig. 1. SEM of single keratinocytes on flat, S1 and S2 substrates.** SEM shows individual well-spread cells adhered to flat surfaces (A,B), S1 substrates with circular micropillars (C,D) and S2 substrates with triangular pillars (E,F). On the S1 substrate, some peripheral pegs are bent (arrows, D), suggestive of localized mechanical forces. Nuclear position varies between cells; whereas some cells exhibit nuclei positioned above the underlying substrates (D,F), others display nuclei positioned within the substrates (C,E). Side views are shown. Scale bars: 10 µm. Data are representative of three independent experiments.

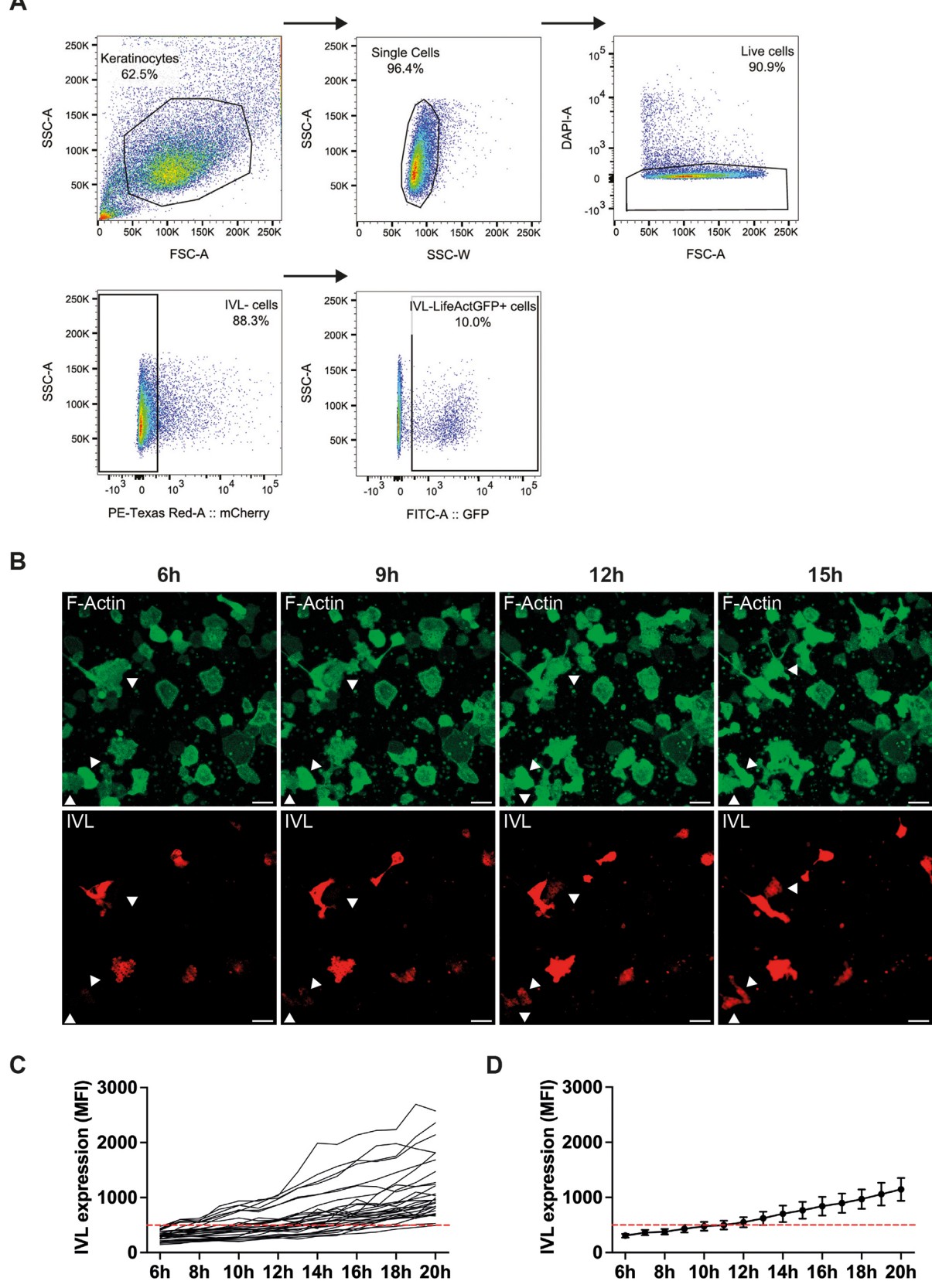

**Fig. 2.** See next page for legend.

independent experiments and subjected to bulk RNA sequencing (RNAseq) (Fig. 3A). In total 18 samples were submitted for RNAseq (three time points with three pooled technical replicates each on each

substrate). Sequencing and bioinformatic analysis were performed by Genewiz, Inc. and are deposited in the Gene Expression Omnibus under the accession code GSE303873.

**Fig. 2. Expression of IVL–mCherry reporter by keratinocytes seeded on S1 culture substrates.** (A) Overview of the FACS sorting strategy to select IVL–mCherry negative cells expressing EGFP (IVL-LifeAct-GFP+ cells) prior to plating on the substrates. Each plot represents a sequential gating step. Arrows indicate the gating sequence. Keratinocytes were first gated by forward (FSC-A) and side scatter (SSC-A), then for single cells and then live cells (DAPI-negative). mCherry-negative (IVL-negative) cells were selected for seeding. (B–D) IVL-LifeAct-GFP+mCherry− cells selected as in A were plated on S1 substrates for 45–60 min and non-adherent cells were washed off. Adherent cells were monitored by live-cell imaging. (B) Representative field of cells. Two cells marked with arrowheads initiated mCherry expression while spread. (C,D) 30 cells that initiated mCherry expression during the imaging period were pooled from three independent experiments for analysis. The IVL–mCherry intensity of each cell was measured by calculating the mean fluorescence intensity (MFI) of cells at different time points. Cells with an MFI>500 were considered to be undergoing differentiation. (C) Traces of individual cells are shown. (D) Mean mCherry signal of all 30 cells. Error bars: 95% c.i. Dashed line shows mean fluorescence of all 30 cells throughout the time course. Scale bars: 50 µm.

Principal component analysis (PCA) revealed that cells on S1 and S2 had highly similar gene expression at 1 h and 4 h, but diverged at 12 h (Fig. 3B). At 1 h and 4 h the differences between cells on S1 and S2 were lower than the differences between different experiments (technical replicates) (Fig. 3B). Nevertheless, cells on S1 and S2 showed significant differences in gene expression at 12 h (Fig. 3B). The majority of differentially expressed genes (100/109 genes, 92%) at 12 h were downregulated on S2 [log2 fold change (LFC) >1 or<−1, adjusted P-value (Padj)<0.05] (Fig. 3C; Table S1). This was also the case when we lowered the LFC threshold to LFC>0.5/LFC<−0.5 (242 out of 337 genes, 72%; Table S1). The full list of differentially expressed genes (LFC>0.5 or LFC<−0.5, Padj<0.05) is found in Table S1.

As shown in Fig. S1, the top GO terms (Subhash and Kanduri, 2016) for genes that were differentially expressed between S1 and S2 at 12 h were 'epidermis development', 'keratinocyte differentiation', 'establishment of skin barrier' and 'peptide crosslinking', which includes genes, such as *TGM1*, that are involved in formation of the epidermal barrier (Steinert and Marekov, 1999; Candi et al., 2005) and genes that are involved in keratinization (a process also linked to differentiation; Redmond and Coulombe, 2021) (Fig. 3D). Thus, the majority of the genes corresponded to genes that are upregulated during keratinocyte terminal differentiation, consistent with our previous findings (Zijl et al., 2019). The top three genes upregulated on S2 at 12 h were *TGFBI*, *COL4A1* and *WNT5A*, all of which are known to be expressed in the basal epidermal layer (Romanowska et al., 2009; Li et al., 2022) (Fig. 3E). *VIM* is expressed at low levels by cultured keratinocytes (Kariniemi et al., 1982) and was upregulated on S2 compared with S1. We previously identified a subset of JUN (JUNB and JUND), FOS (FOS and FOSL1) and MAF (MAF, MAFB, MAFF and MAFG) family AP1 factors that are associated with the differentiation of keratinocytes in suspension (Mishra et al., 2017). We did not see a difference in AP1 expression between cells on S1 and S2, although *FOS* and *MAF* were downregulated at 12 h on both substrates (Fig. 3F).

Real-time quantitative PCR of differentially expressed genes at 12 h confirmed that differentiation was upregulated on S1 and downregulated on S2 compared to the flat substrates, whereas *WNT5A*, *TGFBI* and *VIM* were selectively upregulated on S2 (Fig. S2A,B). As a measure of proliferation, we added 5-ethynyl-2′-deoxyuridine (EdU) to cells cultured on flat, S1 and S2 substrates one hour before fixation. There was no difference in the percentage of EdU+ cells on the different substrates at 12 h (Fig. S3A,B). At 24 h there was a selective decrease in the percentage of EdU+ cells on S1,

correlating with the stimulation of differentiation (Fig. S3A,B). There was no difference in the proportion of apoptotic cells on the different substrates, as visualized by cleaved caspase 3 labelling (Fig. S3C,D). We conclude that stimulation of differentiation on S1 does not involve a distinct gene expression programme.

## Effects of S1 and S2 on cell volume and stiffness
We next compared cell and nuclear volume on the different substrates by confocal microscopy. Keratinocytes were labelled with anti-keratin 14 (K14) and DAPI (Guo et al., 2017; Hansen et al., 2022; Koushki et al., 2023; Li et al., 2021) (Fig. 4A). Cells on S2 had a significantly lower total volume at both 4 h and 12 h (Fig. 4B,C). At 4 h, the nuclear volume was lower and the fold change in nuclear/cytoplasmic ratio was higher on S2, although this was not the case at 12 h (Fig. 4D–G). Given that the cell volume on S1 was similar to that on flat substrates (Fig. 4B,D), the changes in cell volume were specific to S2.

It has previously been reported that a decrease in cell volume can influence cell stiffness and mesenchymal stem cell differentiation (Guo et al., 2017). As actomyosin contractility regulates differentiation on S1 (Zijl et al., 2019) and an increase in cortical tension can increase cell stiffness (Cartagena-Rivera et al., 2016; Harris et al., 2012), we performed atomic force microscopy (AFM) on individual keratinocytes (Fig. 5A) to determine whether differences in cell stiffness correlated with differences in differentiation. Cells on S1 and S2 were significantly softer than cells on flat controls (Fig. 5B–E), ruling out an association between cell stiffness and differentiation on the substrates.

## PEG and DI regulate keratinocyte differentiation
The observation that cells exhibited a reduced cell volume on S2 substrates led us to investigate whether changes in cell volume could directly influence differentiation. We examined the effects of adding polyethylene glycol 300 (PEG300), a molecular crowding compound that decreases cell volume, and deionized water (DI), which increases cell volume (Cai et al., 2019; Guo et al., 2017; Li et al., 2021; Rashid et al., 2023; Tomba et al., 2022). Cells were treated with PBS as controls. Cells were allowed to adhere for 1 h and treated with PEG300, DI or PBS for 3 h prior to fixation and labelling (Fig. 6). We confirmed that PEG300 reduced and DI increased cell volume on flat substrates, relative to the volume of cells treated with PBS (Fig. 6B). PEG300 treatment also led to a small reduction in nuclear volume (Fig. 6C).

Treatment with PEG300 for 24 h reduced differentiation on flat substrates, whereas DI stimulated differentiation (Fig. 7A–C). There was a stronger effect with 5% PEG300 compared to what was seen with 2% PEG300 (Fig. 7D). Treatment with PEG300 also caused a striking change in keratinocyte morphology, with cells becoming more elongated and less spread (Fig. 7B). Moreover, PEG300 inhibited the differentiation of cells in suspension, as demonstrated by western blotting (Fig. 7E; Fig. S4).

Similar effects on differentiation were observed on micropatterned islands, under conditions in which spread area – determined by the micropatterns – did not change (Fig. 8A–C). On the topography substrates, PEG inhibited differentiation on S1 (Fig. 8D), whereas DI promoted differentiation on S2 (Fig. 8E). On all substrates the inhibition of differentiation was reversed by removal of the PEG and the resulting percentage of involucrin-positive cells was greater than in PBS controls (Fig. 8D).

## Potential mechanisms underlying the effects of PEG and DI on differentiation
The Yes-associated protein 1 (YAP1, hereafter YAP) transcriptional regulator plays a key role in integrating mechanical cues from the

Journal of Cell Science

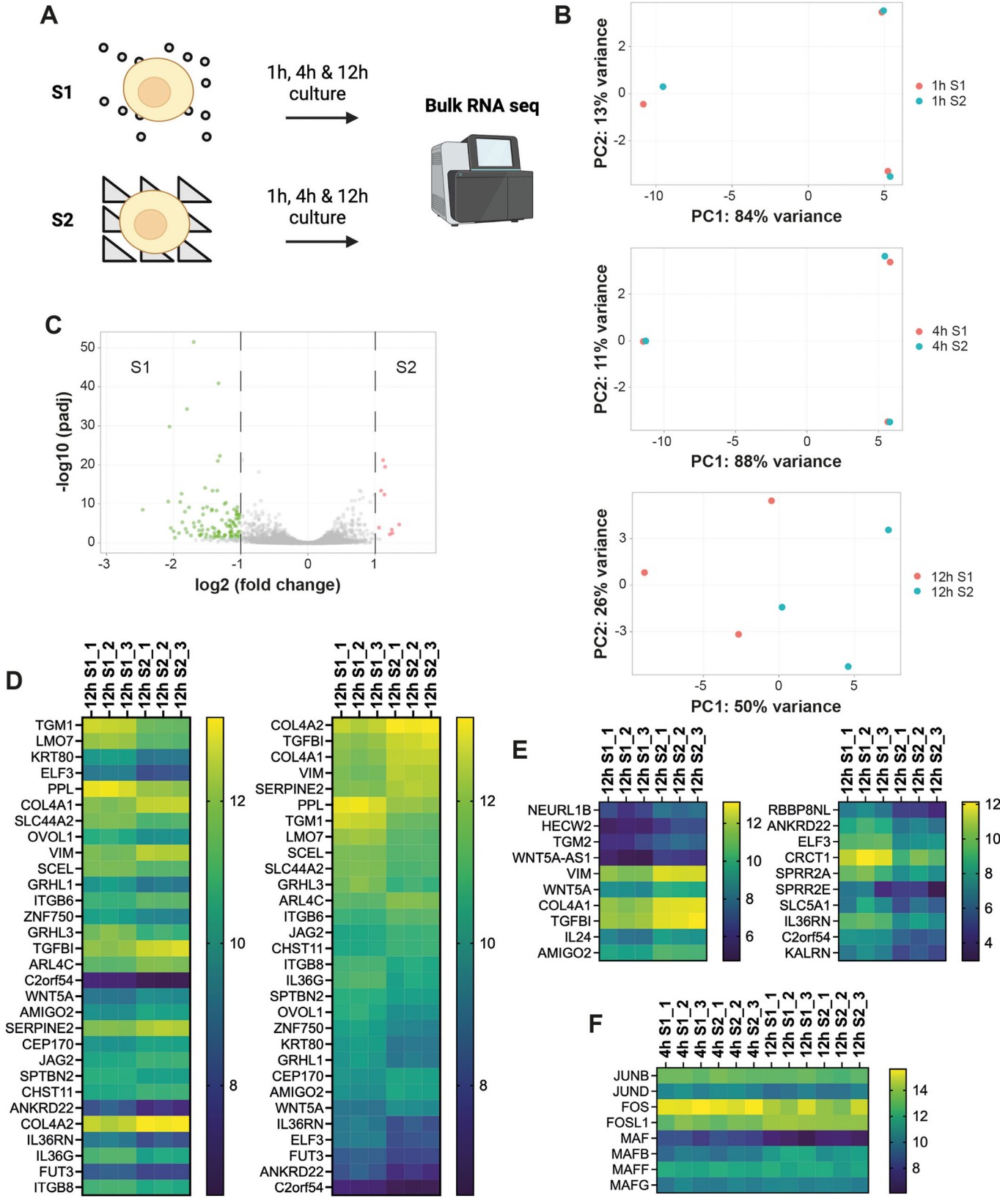

**Fig. 3.** See next page for legend.

cellular microenvironment and eliciting cell responses (Totaro et al., 2017; Walko et al., 2017). YAP is responsive to cell volume changes (Koushki et al., 2023) and is an important fate regulator of epidermal stem cells (Totaro et al., 2017, 2018; Walko et al., 2017). In reconstituted epidermis, YAP activity is controlled by intercellular adhesion (Walko et al., 2017). However, there are

**Fig. 3. Bulk RNA sequencing of keratinocytes seeded on S1 or S2 substrates.** (A) Schematic overview. Cells were cultured for 1 h, 4 h or 12 h in complete FAD medium prior to RNA extraction. RNA was extracted from cells in three independent experiments (replicate experiments are denoted e.g. 1 h S1_1, 1 h S1_2 and 1 h S1_3). Created in BioRender by Zijl, S., 2025. https://BioRender.com/udy2d42. This figure was sublicensed under CC-BY 4.0 terms. (B) Principal component analysis (PCA) plots comparing cells on S1 (red dots) with cells on S2 (blue dots) at different time points. Each data point in the graph represents a different sample. PC, principal component. (C) Volcano plot showing global transcriptional differences between cells on S1 and S2 at 12 h. Each data point represents one gene. The $\log_2$ fold change (LFC) of each gene is shown on the x-axis and the $-\log_{10}$ of its Benjamini–Hochberg adjusted P-value (Padj) is shown on the y-axis. Genes with a Padj<0.05 and LFC>1 (red dots) are upregulated on S2. Genes with Padj<0.05 and LFC<−1 (green dots) are downregulated on S2. (D,E) The most significantly differentially expressed genes at 12 h. (D) Heatmaps of the 30 most significantly differentially expressed genes (DEGs) on S1 and S2 at 12 h. Genes were sorted by their Padj (left-hand column) or expression based on $\log_2$ transformed expression values and LFC in expression (right-hand column). Left-hand column, the most significantly DEGs (lowest Padj) are shown first. Right-hand column, highly expressed genes upregulated on S2 (compared to S1) are shown first, followed by highly expressed genes downregulated on S2. Each square represents the ($\log_2$ transformed) expression level for a different gene (rows) or sample (columns). (E) Heatmaps of the top 10 upregulated genes (highest LFC) (left-hand column) and downregulated genes (highest negative LFC) (right-hand column) on S2 at 12 h (compared to S1). Genes were selected based on a Padj<0.05 and ranked according to their LFC. Each square represents the $\log_2$ transformed expression level for a different gene (rows) or sample (columns). (F) Heatmaps of selected AP1 transcription factors showing changes in expression at 4 h or 12 h relative to 1 h. Each square represents the $\log_2$ transformed expression level for a different gene (rows) or sample (columns).

conflicting reports regarding the role of YAP in keratinocyte differentiation in culture (Connelly et al., 2010; Totaro et al., 2017). On large square micropatterned islands, YAP strongly inhibits differentiation (Totaro et al., 2017), whereas on circular micropatterned islands differentiation is mainly controlled by MAL and SRF (Connelly et al., 2010; Walko et al., 2017). To investigate whether YAP could regulate differentiation in response to cell volume changes, we measured YAP localization on flat, S1 and S2 topographies and in response to PEG300 and DI (Fig. S5). Nuclear YAP was decreased on both S1 and S2, increased in response to PEG treatment and was unaffected by DI. There was therefore no clear correlation between nuclear YAP and differentiation, consistent with our previous work (Walko et al., 2017).

The plasma membrane of cells is made water permeable through the expression of aquaporins (AQPs). In many cell types, changes in intracellular water concentration are associated with changes in intracellular $Ca^{2+}$ concentration (Hoffmann et al., 2009; Jahn et al., 2021). We used our RNAseq dataset to examine expression of aquaporins, ion channels and genes involved in cell volume regulation (Jahn et al., 2021) on cells seeded on S1 and S2 (Fig. 9A,B). AQP3 was the most highly expressed aquaporin on both substrates (Fig. 9A), but expression did not change at the timepoints examined (Fig. 9B). AQP6 expression was lower but upregulated on both S1 and S2 at 4 h and 12 h (Fig. 9B). In contrast, AQP10 was selectively downregulated on S2 at 4 h and 12 h (Fig. 9B). Leucine-rich repeat-containing 8E (LRRC8E), a key component of the volume-regulated anion channel (Trothe et al., 2018), Piezo-type mechanosensitive ion channel component 1 (Piezo-1), solute carrier family 12 member 4 (SLC12A4) and TGF-β stimulated clone-22D2 and -4 (Magaña-Ávila et al., 2025) also showed different changes in expression on S1 and S2 (Fig. 9B). In contrast, transient receptor potential vanilloid-3 (TRPV3), TRPV4, TSC22D1, TSC22D3 and

'with-no-lysine' (WNK) kinases showed similar expression dynamics on S1 and S2 (Fig. 9B). These results suggest there might be different cell volume responses to S1 and S2.

Keratinocytes can respond to changes in their physical environment by changing intracellular ion concentrations. In response to mechanical stretch and deformation, for example, keratinocytes release $Ca^{2+}$ from the endoplasmic reticulum to mitigate the effects of mechanical stress (Nava et al., 2020). To see whether changes in intracellular $Ca^{2+}$ or other ions could be responsible for the effects of DI and PEG on differentiation, we treated cells on flat and S2 substrates with a cell-permeable analogue of 1,2-bis-(2-aminophenoxy)ethane-N,N,N′,N′-tetraacetic acid acetoxymethyl ester (BAPTA-AM), to chelate intracellular $Ca^{2+}$ (Tsien, 1981; Tang et al., 2007; Nava et al., 2020), or with gadolinium ions ($Gd^{3+}$) to inhibit ion channels (predominantly stretch-activated $Ca^{2+}$ channels and transient receptor protein channels; Adding et al., 2001; Bagley et al., 2024). Ion concentrations can be affected by water transport (Cadart et al., 2019), and thus we also treated cells with DFP00173, an inhibitor of aquaporin-3 (AQP3i, Sonntag et al., 2019; de Boer et al., 2023).

Treatment with BAPTA-AM or AQP3i inhibited the effects of DI on differentiation, whereas $Gd^{3+}$ did not (Fig. 9C,D). These treatments, nonetheless, did not reverse the differentiation inhibitory effects of PEG300 (Fig. 9C) and BAPTA-AM did not affect differentiation on S2 (Fig. 9D). These results suggest that the effect of DI in stimulating differentiation might be partly mediated via effects on intracellular $Ca^{2+}$ or water channels. Our results also suggest that the differentiation inhibitory effect of S2 does not depend on changes in intracellular $Ca^{2+}$, arguing against a role for nuclear deformation on S2 (Nava et al., 2020).

## DISCUSSION

As part of our exploration of adhesion-related stem cell niche factors (Fig. 10), we have investigated the response of human epidermal stem cells to topographical features that regulate the differentiation of spread cells (Zijl et al., 2019). We found that cells on the differentiation-inhibiting substrate S2 had a lower volume than cells on the differentiation-promoting substrate S1, prompting us to examine the effect of modulating cell volume with PEG300 and DI. In all contexts tested, a reduction in cell volume inhibited differentiation, whereas an increase in volume-stimulated differentiation. Keratinocyte differentiation has long been known to correlate with an increase in cell size (Sun and Green, 1976; Watt and Green, 1981; Jones and Watt, 1993) but until now evidence for a causative relationship was lacking. Our findings lend support to the hypothesis that reduced substrate contact and increased cell volume can synergize to promote keratinocyte differentiation (Watt, 1988). Cell volume is known to regulate differentiation of mesenchymal stem cells (Guo et al., 2017) and it might also have synergistic effects with cell adhesion in additional cell types (Saraswathibhatla et al., 2023).

Recent studies have demonstrated an association between cell volume and the mechanical properties of the cell cortex and cytoplasm, including cell stiffness (Guo et al., 2017; Jiang and Sun, 2013). Myc-depleted keratinocytes exhibit increased cell stiffness, which correlates with a decrease in cell area, decreased cortical actin (Bernabé-Rubio et al., 2023) and reduced cell size (Zanet et al., 2005). Nevertheless, there was no correlation between cell stiffness and volume in our current studies because bulk stiffness was lower on both S1 and S2 compared to what was seen on flat substrates. We have previously found that cell stiffness is affected by the location of keratinocytes on topographies that mimic the undulations of the

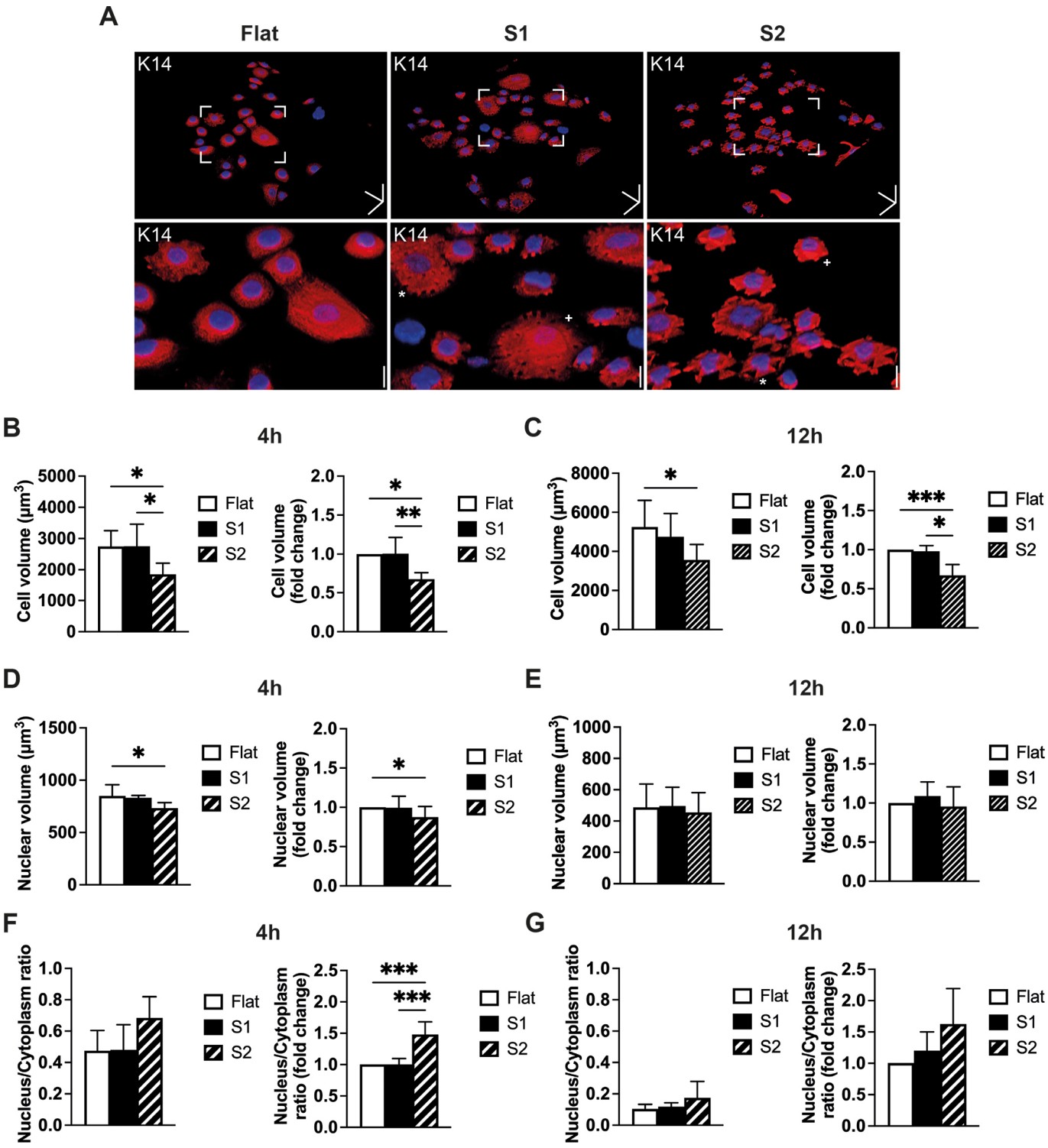

**Fig. 4. Volume of cells seeded on different substrates.** (A) Representative 3D reconstruction images of anti-keratin-14 (K14, red) and DAPI (blue)-labelled cells grown on flat, S1 and S2 substrates for 4 h. Boxes: areas of interest shown at higher magnification below. Scale bars: 50 µm (*x*, *y* and *z* direction), 20 µm (magnified images, *z* direction). Cell (B,C) and nuclear (D,E) volume measurements on flat, S1 and S2 substrates after 4 h (B,D) or 12 h (C,E) of culture. Total and fold change (normalized to flat substrates) in volume are shown. (F,G) Nucleus to cytoplasm ratios at 4 h (F) and 12 h (G) on flat, S1 and S2 substrates; fold changes are relative to flat substrates. Data are from three independent experiments in which >300 cells were analysed per experiment. Three technical replicates were measured per substrate. *$P<0.05$; **$P<0.01$; ***$P<0.001$ [one-way ANOVA plus Tukey's test (absolute volumes); Kruskal–Wallis plus Dunn's test (fold changes)]. Error bars show s.d.

epidermal–dermal junction (Mobasseri et al., 2019), leading us to conclude that there are multiple factors that affect keratinocyte stiffness (Cartagena-Rivera et al., 2016; Harris et al., 2012).

Using bulk RNA sequencing, we did not detect unique features of the terminal differentiation programme in cells responding to S1. Many transcription factors are mechanosensitive

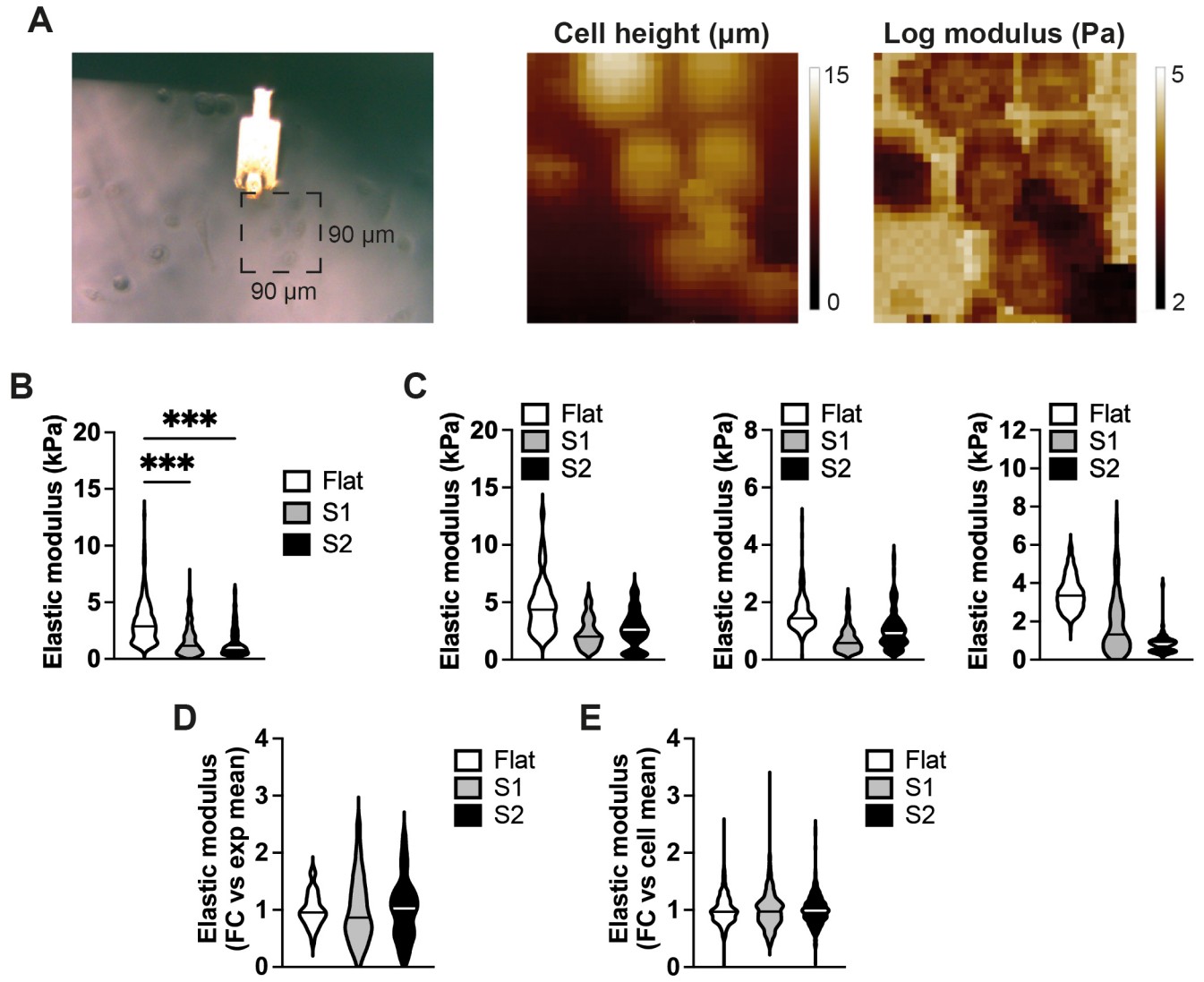

**Fig. 5. Stiffness measurements of cells seeded on different substrates.** (A) Representative brightfield image (left-hand side) of an AFM probe measuring bulk elastic modulus (cell stiffness) of cells on a flat substrate. Height (middle panel) and log modulus (right hand side) measurements are shown. (B) Quantification of stiffness of cells grown on flat, S1 and S2 substrates for 12 h (n=3 independent experiments). Ten randomly chosen cells were analysed per experiment (20 measurements per cell). (C) Data from the individual experiments shown in B. (D) Variation in cell stiffness within individual experiments shown in C. Values normalized to the mean (1.0) of individual experiments. (E) Variation in stiffness between individual cells shown in C. Values normalized to the mean (1.0) of individual cells. Plots in B–E show the median (horizonal line) and range in values. FC, fold change; kPa, kilopascal. ***$P<0.001$ (Kruskal–Wallis test plus Dunn's correction).

(Bao et al., 2019; Dupont and Wickström, 2022), including SRF, MAL, and YAP and TAZ (TAZ is also known as WWTR1), which are known regulators of epidermal differentiation (Zijl et al., 2022). The different effects of S1 and S2 on differentiation did not correlate with YAP nuclear location, although nuclear YAP increased in response to PEG300 (Koushki et al., 2023).

Cells are known to be densely packed with nucleic acids, proteins, lipids and other macromolecules, and this molecular crowding influences many aspects of cellular biology, including cytoskeletal organization and cell adhesion (Subramanya and Boyd-Shiwarski, 2024). We have previously observed differences in nuclear morphology and cytoskeletal organization in keratinocytes seeded on S1 and S2 substrates (Zijl et al., 2019), and changes in cell volume are known to affect molecular crowding (Subramanya and Boyd-Shiwarski, 2024). This leads us to speculate that crowding might play

a role in volume-regulated epidermal differentiation. Although more work needs to be done to explore the underlying mechanisms, it will be interesting to start with the genes that are differentially regulated on S2 at 12 h, such as vimentin (Wu et al., 2022), *SLC44A2* and the microtubule-associated genes *ARL4C* and *CEP170* (Wei et al., 2009; Pillai et al., 2015).

Cell volume regulation is an integral part of normal physiological processes within tissues (Hoffmann et al., 2009). Epidermal cells, like other cell types, can actively counteract swelling or shrinkage via regulatory cell volume changes (Jahn et al., 2021). Cell volume is controlled by multiple ion channels and transporters, several of which are expressed in the epidermis (Jahn et al., 2021). Using bulk RNAseq we catalogued expression of genes linked to cell volume regulation in keratinocytes on S1 and S2 substrates and demonstrated that the membrane-permeable $Ca^{2+}$ chelator BAPTA-AM and an aquaporin 3 inhibitor blocked DI-induced

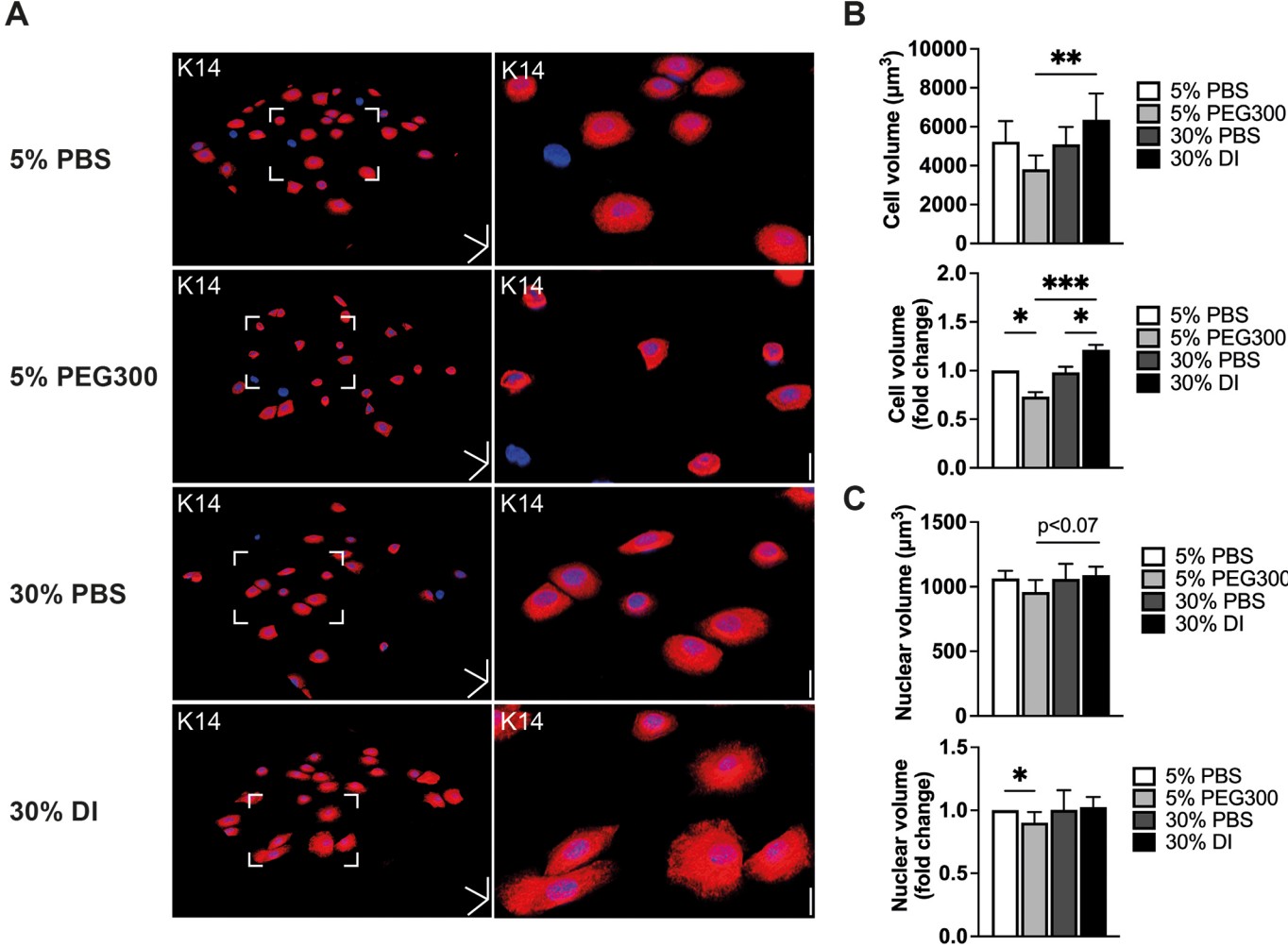

**Fig. 6. Modulation of cell volume on flat substrates.** (A) Representative 3D reconstruction images of cells labelled with DAPI (blue) and anti-keratin 14 (K14) after treatment with 5% PBS, 5% PEG300, 30% PBS or 30% DI for 3 h. Boxed areas of interest in left hand panels are enlarged in right hand panels. Scale bars: 50 µm (*x*, *y* and *z* direction), 20 µm (magnified images, *z* direction). (B) Quantification of cell volume and fold change in cell volume (normalized to 5% PBS). (C) Quantification of nuclear volume and fold change in nuclear volume (normalized to 5% PBS). Data from three independent experiments in which >300 cells were analysed per experiment. In each experiment three technical replicates were analysed per condition. *$P<0.05$, **$P<0.01$. ***$P<0.001$ [one-way ANOVA plus Šídák's) test (volume measurements); Kruskal–Wallis plus Dunn's test (fold changes)]. Error bars show s.d. PBS, phosphate-buffered saline; PEG300, polyethylene glycol 300; DI, deionized water.

differentiation. Further analysis of the roles of specific volume regulatory genes within the epidermis will clearly be of interest.

It will be intriguing to discover the extent to which cell volume regulation of differentiation involves parameters such as nuclear confinement and actin tension. Actin organization differs between cells plated on S1 and S2 substrates (Zijl et al., 2019), and there is an increased concentration of F-actin around the pillars of topography S1, resembling podosomes (Rafiq et al., 2019). Aquaporins are associated with a variety of cell adhesion receptors expressed by keratinocytes, such as α5β1 integrin and desmoglein-2 (Smith and Stroka, 2023), and thus it will be interesting to investigate whether they are involved in the cell volume response. Cell volume might affect different cellular mechanisms (Roffay et al., 2021; Li et al., 2021) that could independently or synergistically regulate epidermal differentiation. As the interplay between different cellular components and different differentiation stimuli (Fig. 10) becomes clearer, we will be able to build a comprehensive map of the different mechanisms by which the stem cell niche regulates epidermal cell fate decisions.

## MATERIALS AND METHODS
### Keratinocyte culture
Primary human keratinocytes (strains km, kn or kp) from neonatal foreskin (see below) were cultured on a feeder layer of mitotically inactivated J2 3T3 cells (see below) in one part Ham's F12 medium (F12; Gibco) and three parts Dulbecco's modified Eagle's medium (DMEM; Gibco) supplemented with $1.8 \times 10^{-4}$ M adenine (basal FAD medium; Sigma), as described previously (Walko et al., 2017; Vietri Rudan et al., 2024). Basal FAD medium was supplemented with 10% fetal calf serum (FCS; Gibco), 0.5 µg/ml hydrocortisone, 5 µg/ml insulin, $10^{-10}$ M cholera enterotoxin and 10 ng/ml epidermal growth factor (Peprotech) to make complete FAD medium. Keratinocytes were isolated by Mr Simon Broad (Cambridge University and King's College London, UK) and used at passage 3–7. J2-3T3 cells were cultured in DMEM containing 10% donor calf serum (Gibco). J2-3T3 cells were isolated by Dr James Rheinwald (Harvard University) and passaged 1–12 times from stocks provided by Dr Rheinwald. In some experiments, 10 µM 5-ethynyl-2′-deoxyuridine (EdU; Click-iT EdU Imaging Kit, Thermo Fisher Scientific) was added to the medium for 1 h prior to fixation. Cells were routinely tested for mycoplasma and were found to be negative.

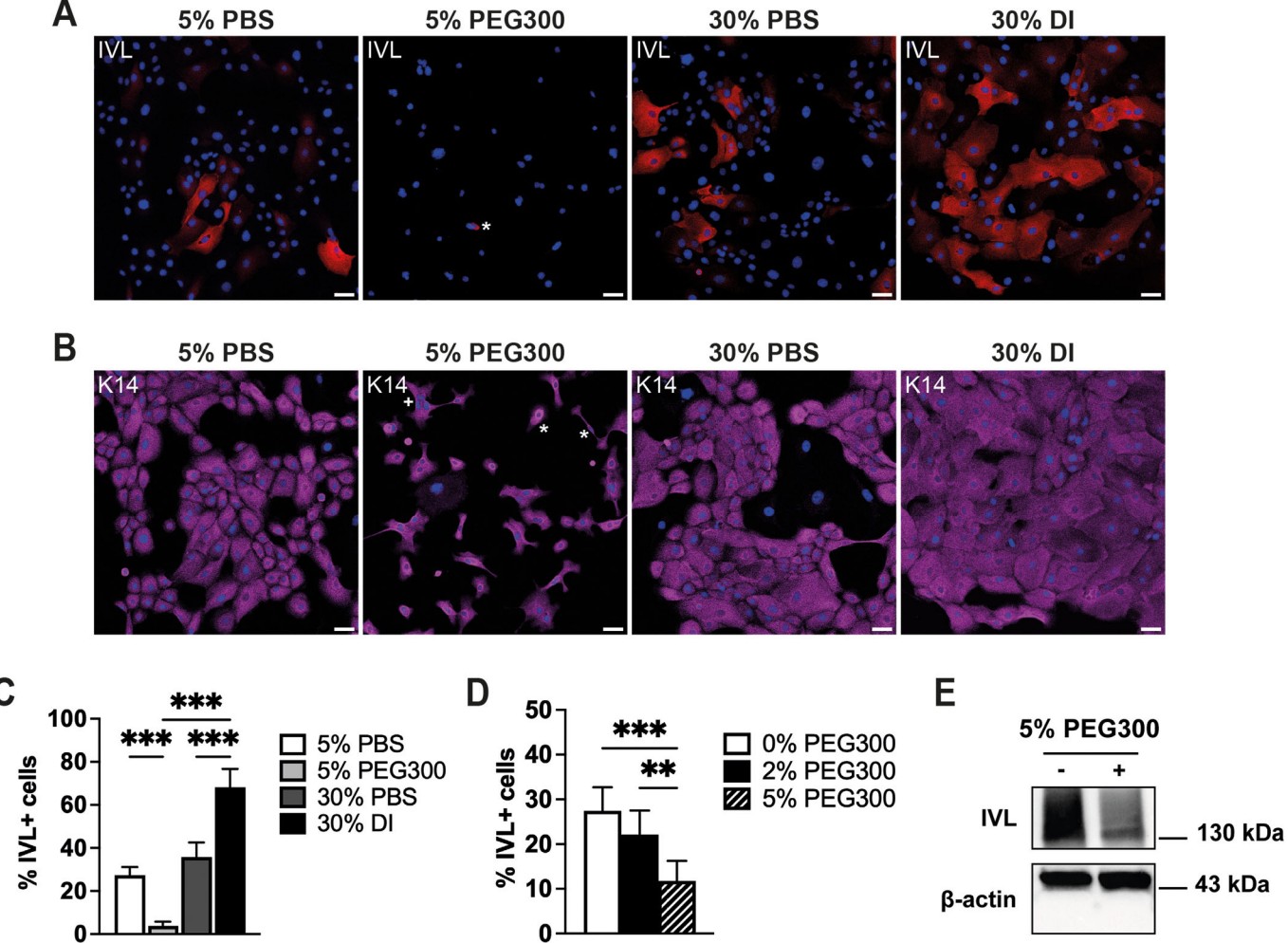

**Fig. 7. Modulation of differentiation of adherent and suspended keratinocytes with PEG300 and DI.** (A,B) Representative images of anti-involucrin (A; IVL, red)- and anti-keratin 14 (B; magenta)-labelled cells with DAPI counterstain (blue). Cells were grown for 24 h on flat substrates (unpatterned 96-well plates) in the presence of 5% PBS, 5% PEG300, 30% PBS or 30% DI. Scale bars: 50 μm. Asterisks, a rare differentiated cell (A); examples of cells with altered morphology (B). (C) Quantification of the percentage of differentiated cells (% IVL+ cells) of cells grown under the conditions shown in A, B. (D) Quantification of the percentage of differentiated cells (% IVL+ cells) on flat substrates in the presence of different concentrations of PEG300. Cells were grown for 24 h. In C and D, data are from three experiments in which >1000 cells were analysed per experiment. Experiments were performed with three technical replicates per condition. **$P<0.01$. ***$P<0.001$ (one-way ANOVA plus Tukey's test). Error bars show s.d. (E) Western blot of keratinocytes cultured in suspension for 24 h in the presence of 5% PBS (−) or 5% PEG300 (+). Cell lysates were probed with antibodies against involucrin (IVL) to visualize differentiation; antibodies against β-actin were used as a loading control. Full blot and replicates are shown in Fig. S4.

## Polystyrene topographies

A silicon (Si) wafer template was fabricated using photolithography and etching, as described previously (Zijl et al., 2019; Kelvin Nanotechnology Ltd, UK). The wafer was coated with PDMS (Sylgard 184; Qin et al., 2010) overnight at room temperature and then cured by heating at 80°C for at least 5 h. The cured PDMS was cooled to room temperature and then coated with 25% polystyrene (Goodfellow) dissolved in gamma-butyrolactone (GBL, Acros Organics; Wang et al., 2011). GBL was evaporated for 4 h at 95°C, followed by >12 h at 150°C. To create flat control substrates, the same process was performed using flat PDMS. After oxygen plasma treatment (1–2 min at 0.3–0.4 mBar) (Jokinen et al., 2012) and collagen coating (rat tail collagen type I, 20 μg/ml, overnight at room temperature; Corning), the polystyrene surfaces were used for cell seeding. Keratinocytes were suspended in complete FAD medium and seeded at a density of $7.5\times10^4$ cells per cm².

## Scanning electron microscopy

Keratinocytes were grown on collagen-coated polystyrene substrates for 24 h and chemically fixed in 2.5% (w/v) glutaraldehyde (Science Services, EM grade, E16200) in 0.1 M PHEM buffer. Briefly, 2× fixative was added

to the cells in a 1:1 ratio with medium and incubated for 15 min at room temperature. The 2× fixative was then replaced with fresh 1× fixative, and the cells were incubated for an additional hour at room temperature. The cells were washed twice with 0.1 M PHEM buffer and transferred to new wells to remove residual traces of glutaraldehyde. Further fixation was continued with two washes in 0.1 M sodium cacodylate buffer. The samples were post-fixed in freshly prepared 1% (w/v) osmium tetroxide (Science Services, E19152), 0.8% potassium-ferrocyanide (Merck, 4984) in 0.1 M sodium cacodylate buffer for 45 min, protected from light and rinsed four times with distilled (d)H₂O. Afterwards, samples were incubated with 1% (w/v) uranyl acetate (Serva, 77870.01) in dH₂O for 30 min on ice and, subsequently, serially dehydrated in increasing concentrations of cold ethanol on a shaker (3×25%, 2×50%, 2×70% and 2×90%). The samples were then washed twice with 100% ethanol for 10 min at room temperature, critical point dried (Leica Microsystems) and mounted on clean SEM stubs with carbon-stickers. The stubs were gold-sputter coated to increase their conductivity (Quorum, QR150 S). Samples were imaged with a Zeiss 540 Crossbeam at 5 kV acceleration voltage, a 700 pA current, and a secondary electron detector at 0° (top view) and 54° tilt (side view) respectively.

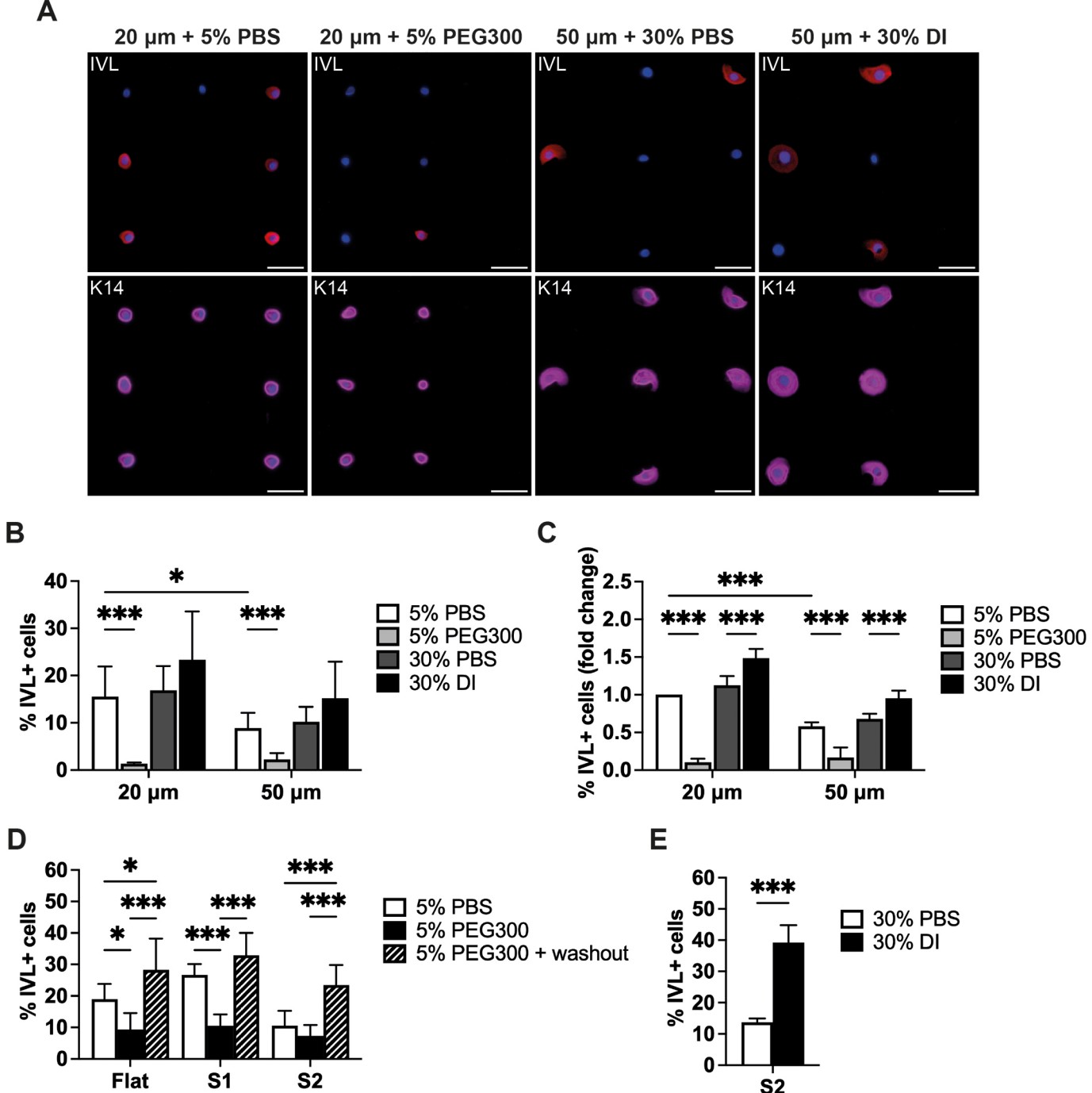

Fig. 8. Effects of PEG300 and DI on cells on micropatterned islands and S1 and S2 substrates. (A) Representative images of anti-involucrin (IVL, red) and anti-keratin14 (K14, magenta) staining of cells grown on 20 µm and 50 µm diameter micropatterned islands. Cells were counterstained with DAPI (blue). Cells were grown in the presence of 5% PBS, 5% PEG300, 30% PBS or 30% DI for 24 h. Scale bars: 50 µm. (B,C) Quantification of (B) the percentage of differentiated cells (% IVL+ cells) and (C) the fold change in the percentage of differentiated cells (% IVL+ cells, normalized to 20 µm islands treated with 5% PBS). Results are from three independent experiments performed with three technical replicates per condition. (D) Quantification of the percentage of differentiated cells (% IVL+ cells) on flat, S1 and S2 substrates. Cells were cultured in the presence of 5% PEG300 for 24 h and fixed or allowed to recover in complete FAD medium for 24 h (washout). (E) Cells grown on S2 for 24 h in the presence of 30% PBS or 30% DI. More than 1000 cells were analysed per experiment. *$P<0.05$; *** $P<0.001$ [two-way ANOVA lpus Šídák's) test (B,C); one-way ANOVA plus Tukey's test (D,E)]. Error bar show s.d.

## Lentiviral infection

The SV40-puro cassette comprising SV40 promoter and puromycin resistance gene sequences in pLenti-INV3700-mCherry as altered to include the hPGK-LifeAct-EGFP from pLenti.PGK.LifeAct-GFP.W and a WPRE element (Hiratsuka et al., 2020) was replaced with the hPGK-LifeAct-EGFP-WPRE sequence from pLenti.PGK.LifeAct-GFP.W (Addgene #51010; Belin et al., 2014) using the In-Fusion HD Cloning Kit (Takara). DNA fragments were obtained by PCR amplification. The sequence of the resulting plasmid (pLenti-INV-mCherry-LifeAct-EGFP) was validated by Sanger sequencing (Source Bioscience Sequencing, Cambridge). Plasmids and sequences can be provided upon request.

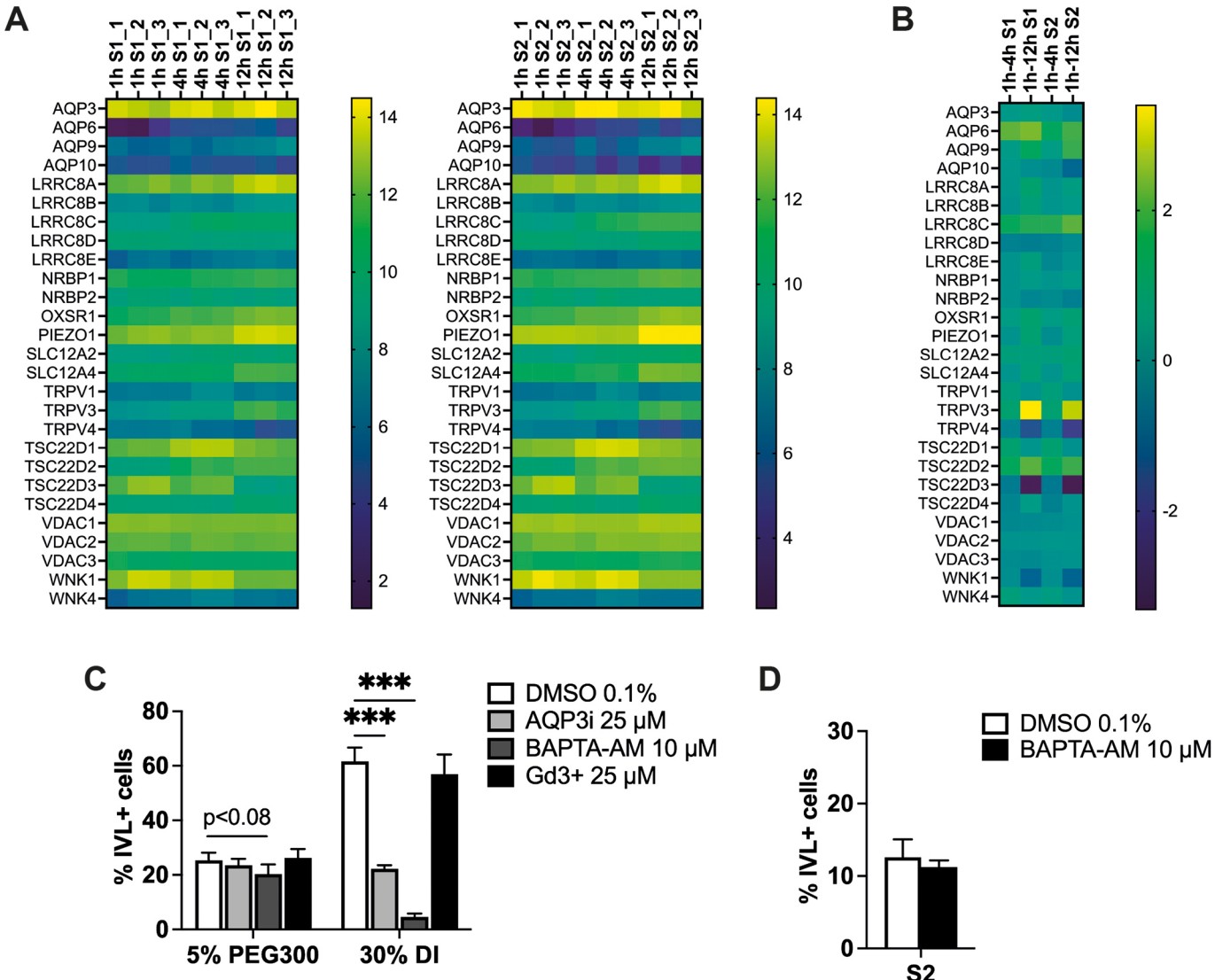

**Fig. 9. Potential role of aquaporins, intracellular Ca²⁺ ions and ion channels in regulating differentiation.** (A,B) Heatmaps of the expression of aquaporins, ion channels and genes involved in cell volume control, plotting relative expression levels in cells seeded on S1 and S2 for 1 h, 4 h and 12 h (A) and relative changes between 1 and 4 h or 1 and 12 h (B). Each square represents the $\log_2$ transformed expression level for a different gene (rows) or sample (columns). (C,D) Quantification of the percentage of differentiated cells (% IVL+ cells). Cells were grown on flat (C) or S2 substrates (D) for 24 h in the presence of DMSO (0.1%) or BAPTA-AM (10 μM) (C,D) or 25 μM $Gd^{3+}$ (C) and the percentage of differentiated cells was quantified (% IVL+ cells). Results are from three independent experiments performed with three technical replicates per condition. More than 500 cells were analysed per experiment. ***$P<0.001$ [two-way ANOVA plus Dunnett's test (C); two-tailed unpaired $t$-test (D)]. Error bars show s.d. MFI, mean fluorescence intensity.

### Flow cytometry

Single-cell suspensions of keratinocytes transduced with pLenti-IVL-mCherry-LifeAct-EGFP were suspended in Phenol Red-free complete FAD medium, labelled with DAPI and flow sorted based on the expression of LifeAct–EGFP and the absence of IVL–mCherry. Sorting was performed on a FACSAria III Cell Sorter (BD Biosciences). Flow data were analysed using FlowJo (BD Biosciences, version 10). Uninfected cells were used as a negative control to set the EGFP+ and mCherry– gates. DAPI was used to exclude dead cells. After sorting, cells were resuspended in pre-warmed complete FAD medium and plated on a J2 feeder layer. When the cultures had reached ∼80% confluence, they were harvested for live-cell imaging.

### Live-cell imaging

pLenti-LifeAct-EGFP-INV-mCherry-infected keratinocytes were disaggregated and seeded on polystyrene (PS) topographies for 45–60 min. Non-adherent cells were removed, and adherent cells were incubated at 37°C for 4–6 h before being transferred to Phenol Red-free

FAD medium supplemented with 1 mM sodium pyruvate, 36.5 mM sodium bicarbonate (Gibco) and 100 mM HEPES. Imaging was performed on an upright A1R multi-photon microscope (Nikon) at 37°C. Cells were imaged at 1 h intervals. Images were acquired with multiple $z$-slices ($z$-interval: 5–10 μm step size) and combined in maximum projections. The resolution of the images was 512×512 pixels and the magnification was 25×. Cells with an mCherry mean fluorescent intensity (MFI) of >500 were scored as differentiating cells (active IVL promoter) (Hiratsuka et al., 2020).

### Immunofluorescence staining

To label cells with antibodies to involucrin (SY3 or SY7 mouse monoclonal antibodies; Hudson et al., 1992), cleaved caspase-3 (rabbit polyclonal antibody; Cell Signaling Technology 9664), YAP (rabbit polyclonal antibody; Cell Signaling Technology 14074) or keratin 14 (chicken polyclonal antibody; Biolegend 906001, 906004), cells were first fixed in 4% (w/v) paraformaldehyde (PFA, Sigma) diluted in phosphate-buffered saline (PBS, Sigma) for 15 min at room temperature then permeabilized in 0.2% (w/v) Triton X-100 (Tx100, Sigma, diluted in PBS for 15–20 min at

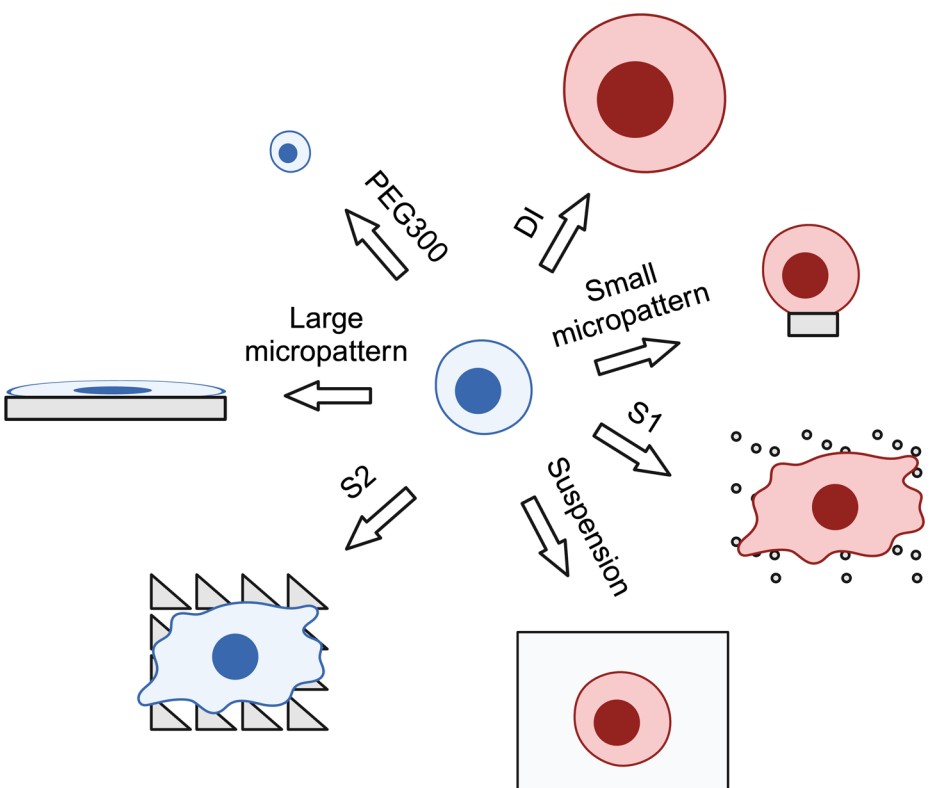

**Fig. 10. Overview schematic.** Cultured epidermal stem cells (blue) can be induced to differentiate (pink) by seeding in suspension, on small micropatterned islands, on S1 substrates or by treatment with DI. The cells remain undifferentiated on S2 substrates, large micropatterned islands or when treated with PEG300. Created in BioRender by Zijl, S., 2025. https://BioRender.com/k4xpug4. This figure was sublicensed under CC-BY 4.0 terms.

room temperature (RT). Samples were washed three times with PBS and blocked in PBS containing 10% (v/v) FBS and 0.25% (w/v) cold water fish skin gelatin (Sigma) (Connelly et al., 2010) for 1 h at room temperature. Primary antibodies were diluted in blocking buffer and incubated with the fixed cells overnight at 4°C or for 1 h at RT. After washing 3× with PBS cells were incubated with phalloidin (Invitrogen A12379), conjugated secondary antibodies (Invitrogen) and 4′,6-diamidino-2-phenylindole (DAPI, Invitrogen D1306, 1 µg/ml) in blocking buffer for 1 h at room temperature in the dark. After staining, samples were washed three times with PBS and mounted onto glass six-well plates (Cellvis) or Superfrost Plus slides (Thermo Fisher Scientific) with ProLong Gold Antifade Mountant (ThermoFisher) or Mowiol 4-88 (Sigma). Details of antibodies and other staining reagents can be found in Table S2.

### Confocal microscopy

Immunofluorescence labelled cells on polystyrene substrates were analysed by confocal microscopy. Images taken with 10×, 20× or 40× magnification objectives were acquired using dry objectives on an A1 upright confocal (Nikon) at the Nikon Imaging Centre (King's College London). Images taken with a 63× magnification objective were acquired on an A1R inverted confocal (Nikon), using an oil objective (immersion oil, Nikon). Confocal images were analysed using ImageJ/Fiji software. Mean fluorescence intensity (MFI) values were computed using an automated imaging pipeline and were constructed using Jython (Python implementation in ImageJ/Fiji), as described previously (Louis et al., 2022). Pipelines were used to compute the MFI values and morphological measurements of individual cells. Subsequent data analysis was done in ImageJ/Fiji and Excel software. Unless otherwise stated, intensity values represent MFI values per cell. The MFI is the average intensity of all cells in an experiment. To determine the percentage of positive cells for a marker of interest, intensity thresholds were set based on the background intensity (areas in the images without staining for the marker of interest) and cells labelled with a secondary antibody only control. Unless otherwise stated, cells were considered differentiated (IVL+) if their cytoplasmic MFI for IVL was higher than that of cells stained with a secondary antibody control (cells that were not stained with a primary antibody against IVL, but only with a secondary antibody).

IVL+ cells also had MFIs higher than the background intensity (MFI of areas negative for IVL). Image analysis pipelines can be provided upon request.

### Cell volume measurements

To quantify the volume of cells grown on polystyrene substrates, cells were fixed with 3.7% formaldehyde or 4% PFA in PBS, labelled with anti-keratin 14 to mark the cytoplasm and DAPI to mark nuclei, and imaged by confocal microscopy. The $z$-slice interval was set at 0.5 µm (Hansen et al., 2022; Koushki et al., 2023). Confocal measurements of cell volume have been shown to be similar to super resolution imaging measurements (Guo et al., 2017; Lee et al., 2019, 2021). Z-stacks were deconvoluted using the NIS elements imaging software (Nikon), and cell and nuclear volumes were calculated in ImageJ/Fiji (NIH, voxel counter plug-in) (Koushki et al., 2023). Thresholding (default method) was performed to exclude background staining. Total thresholded volumes were divided by the number of cells per image, to obtain the average cell/nuclear volume per cell for each image. Averages for different images were combined to create averages for different experiments.

### Cell volume manipulation

Cells were treated with 5% (v/v) PEG300 (Sigma) or 30% (v/v) DI (Cai et al., 2019; Guo et al., 2017; Li et al., 2021; Rashid et al., 2023; Tomba et al., 2022). Cells treated with 5% (v/v) PBS (for 5% PEG300) and 30% (v/v) PBS (for 30% DI) were used as controls. For cell volume manipulation on PS substrates, cells were allowed to attach for 45–60 min and non-adherent cells were removed. Afterwards, the medium was switched to FAD medium containing 5% PBS, 5% PEG300, 30% PBS or 30% DI. Cells were cultured for a further 24 h to allow differentiation to occur (Connelly et al., 2010) and fixed and stained for analysis. For cell volume manipulation on unpatterned substrates, cells were seeded on collagen-I coated (20 µg/ml in PBS) 96-well plates (Greiner, µClear black) at a density of $1.0 \times 10^4$–$1.5 \times 10^4$ cells/well ($3.0 \times 10^4$–$5.0 \times 10^4$ cells/cm²) for 45–60 min. After removal of non-adherent cells, the cells were transferred to complete FAD containing 5% PBS, 5% PEG300, 30% PBS or 30% DI and cultured for 24 h before fixation and labelling.

## Pharmacological inhibitors

To chelate intracellular $Ca^{2+}$, cells were treated with a cell-permeable analogue of 1,2-bis-(2-aminophenoxy)ethane-N,N,N′,N′-tetraacetic acid acetoxymethyl ester (BAPTA-AM, Abcam, 10 μM) (Tsien, 1981; Tang et al., 2007; Nava et al., 2020). Gadolinium trichloride ($Gd^{3+}$, Sigma, 25 μM) was used to block ion channels (predominantly stretch-activated $Ca^{2+}$ channels and transient receptor protein channels, but also others; Adding et al., 2001; Bagley et al., 2024). DFP00173 (AQP3i, Cambridge Bioscience, 25 μM) was used to inhibit water transport through aquaporin-3 (Sonntag et al., 2019; de Boer et al., 2023). All drugs were diluted in dimethyl sulfoxide (DMSO, Sigma). Cells treated with DMSO only (vehicle) were used as negative controls (maximum final concentration of DMSO, 0.1%).

## Differentiation on micro-patterned islands and in suspension

20 μm and 50 μm diameter micropatterned islands were prepared using custom Quartz photomasks (JD Photodata) as described previously (Louis et al., 2022). To induce terminal differentiation in suspension, pre confluent keratinocytes were disaggregated and resuspended at a concentration of $10^5$ cells per ml in complete FAD medium supplemented with 1.45% methylcellulose (4000 centipoises, Aldrich). The cell suspensions were then plated in 35-mm-diameter bacteriological plastic dishes coated with 0.4% polyHEMA (Sigma). After incubation for up to 24 h at 37°C the methylcellulose was diluted with PBS and the cells recovered by centrifugation (Louis et al., 2022).

## RNA sequencing and RT-qPCR

RNA isolation was performed using the RNeasy mini kit (Qiagen), according to the manufacturer's instructions. For real-time quantitative PCR (RT-qPCR), complementary DNA (cDNA) was synthesized using the Quantitect Reverse Transcription kit (Qiagen), according to the manufacturer's instructions. The cDNA was mixed with Fast SYBR Green (SYBR, Applied Biosystems), nuclease-free water and custom-made oligonucleotide primers (Sigma). Reactions were performed in 384-well PCR plates (Bio-Rad) using a CFX384 Touch Real-Time PCR Detection System (Bio-Rad). The primers used for qPCR are listed in Table S3. Each sample was run with four technical replicates and three experimental replicates per condition. Gene expression was quantified using the 2−ΔΔCT method (Livak and Schmittgen, 2001). Cycle threshold (Ct) values were averaged across the different technical replicates. Ct values were normalized to the expression of housekeeping genes (the average of *18S*, *GAPDH* and *TBP*). Gene expression in different samples was normalized to control conditions (flat substrates). Melting curves for the different primers were checked after each experiment. Primers that generated unusual melting curves (e.g. melting curves with several peaks) were replaced. Primers that generated Ct values in negative control reactions (nuclease-free $H_2O$ instead of cDNA) were also replaced.

To prepare keratinocytes for bulk RNA sequencing (RNAseq), RNA was isolated using the RNeasy mini kit (Qiagen). Each experiment was performed with three technical replicates per condition. Technical replicates were pooled together. The quality of the RNA was checked on a Qubit 2.0 Fluorometer (Thermo Fisher Scientific). Samples submitted for sequencing had an RNA integrity number of 9.9–10.0. mRNA selection, library preparation and sequencing were performed by GENEWIZ Inc. as described previously (Cujba et al., 2022) according to standardized Illumina protocols. High throughput RNAseq was performed on a NovaSeq 6000 Sequencing System (Illumina). At least $2.0 \times 10^7$ paired-end reads of 150 bp were obtained per sample. The mean quality score of the samples was >35, as determined by FastQC.

Raw RNA sequencing files were processed by GENEWIZ. To remove possible adapter sequences and poor-quality nucleotides, sequencing reads were trimmed with Trimmomatic v.0.36 (Bolger et al., 2014). Quality control was performed using FastQC/0.11.8 (https://www.bioinformatics.babraham.ac.uk/projects/fastqc/). Trimmed reads were mapped to the human GRCh38 reference genome, using STAR aligner v.2.5.2b (Dobin and Gingeras, 2015). Transcript abundance was calculated using 'featureCounts' from the 'Subread' package v.1.5.2 (Liao et al., 2013). Only unique reads that fell within exon regions were counted. Gene expression analysis was performed in R version 3.5.5, using 'DESeq2' (Love et al., 2014). The Wald test was used to generate *P*-values and $\log_2$

fold changes (LFCs). Genes with a Benjamini and Hochberg adjusted *P*-value (*P*adj) <0.05 and an absolute LFC >1 (or <−1) were considered significantly differentially expressed. Volcano plots were generated using the 'EnhancedVolcano' package (doi:10.18129/B9.bioc.EnhancedVolcano). Heatmaps were made using the heatmap.2 function from the 'gplots' package (https://cran.r-project.org/web/packages/gplots/index.html) and plotted by the authors in GraphPad Prism. Principal component analysis (PCA) was performed using the plotPCA function in DESeq2.

## Atomic force microscopy

Keratinocytes were cultured on PS substrates for 12 h and then transferred to FAD medium containing 100 mM HEPES. Measurements were performed on a BioScope Resolve BioAFM (Bruker), which was coupled to an optical microscope (Nikon Eclipse Ti-U). Measurements were carried out on live cells at 37°C using a spherical nitride tip (5 μm) and nitride cantilever (SAA-SPH-5UM, Bruker). For each sample, 24×24 force extension measurements were performed to probe stiffness. Each measurement involved a 10 μm piezo-electric excursion up to a maximum force of 10 nN. The Young's modulus (cell stiffness) was calculated by fitting the force–extension curves with a Hertzian model (spherical), assuming a Poisson's ratio of 0.5, using the Nanoscope software (version 1.8, Bruker). Only the region between 30% and 70% of the maximum force was employed for fitting. For each experiment, ten randomly selected cells were imaged per condition (flat, S1 and S2). 20 force curves were generated per cell. Experiments were repeated three times.

## Western blotting

Cells were lysed for 30 min on ice in RIPA buffer (Sigma) containing Pierce protease inhibitor (Thermo Fisher Scientific), then sonicated for 3×25 s at 4°C (CamSonix C080T Ultrasonic bath, Camlab). Lysates were centrifuged at 16,000 *g* for 10 min at 4°C and the pellets discarded. The protein concentrations of supernatants were determined using a Pierce BCA protein assay kit (Thermo Fisher Scientific), according to the manufacturer's instructions.

For western blotting, 25 μg of protein was diluted in Laemmli SDS sample buffer (Thermo Fisher Scientific), boiled at 95°C for 5 min and loaded into each lane of a 4–15% Mini-PROTEAN TGX Stain-Free precast gel (Bio-Rad). Colour Prestained Protein ladder (New England Biolabs) was loaded in some wells in order to determine the molecular mass of proteins in the experimental samples. Gels were submerged in 1× SDS running buffer (Bio-Rad) and subjected to SDS-PAGE for ∼2 h at 100–120 V. Afterwards, gels were transferred to PVDF membranes in Trans-Blot Turbo Mini 0.2 μm PVDF Transfer Packs (Bio-Rad) using the Trans-Blot Turbo Transfer System (Bio-Rad, 7 mins at 20 V).

Membranes were blocked for 1 h at room temperature in Tris-buffered saline containing 0.025% Tween-20 (0.025% TBST, Severn Biotech) and 5% non-fat skimmed milk (Tesco) then incubated with primary antibodies diluted in blocking buffer overnight at 4°C (Table S2). After washing three times with 0.025% TBST for 10 min membranes were incubated with secondary antibodies conjugated to horseradish peroxidase in blocking buffer overnight at 4°C. Following 3×10 min washes in 0.025% TBST, the blots were developed using Clarity Western ECL Substrate (Bio-Rad), according to the manufacturer's instructions. Protein bands were visualized on a ChemiDoc Touch Imaging System (Bio-Rad). The following antibodies were used: SY7 mouse anti-involucrin (Hudson et al., 1992), HRP-conjugated mouse anti-β-actin (Santa Cruz Biotechnology, sc-47778 HRP) and HRP-conjugated horse anti-mouse IgG (Cell Signaling Technology 7076).

## Statistics and reproducibility

Graphing was performed in Excel or Graphpad Prism (Dotmatics). Statistical analyses were performed in GraphPad Prism. Multiple testing corrections were used where indicated, e.g. one-way ANOVA [+ Tukey's] test means a Tukey's multiple comparisons test was used in combination with a one-way ANOVA test. Only significant ($P<0.05$) and near significant *P*-values ($P<0.10$) are shown. Error bars represent the standard deviation (s.d.) from the mean (mean+s.d.) unless otherwise stated. Images are representative of the indicated number of experiments. Generally, experiments were performed with three experimental replicates per

Journal of Cell Science

condition per substrate and three technical replicates per condition (e.g. per time point), unless otherwise stated.

## Acknowledgements
F.M.W. gratefully acknowledges financial support from EMBO and technical assistance from the EMBL Electron Microscopy Core Facility, the King's College London Nikon Imaging Centre and the King's College London Flow Cytometry Core Facility. S.G.-M. was supported in part by the Francis Crick Institute, which receives its core funding from Cancer Research UK (CC0102), the UK Medical Research Council (CC0102) and the Wellcome Trust (CC0102).

## Competing interests
F.M.W. is EMBO Director, and a director of Fibrodyne Ltd. A.M. is an employee of CN Bio.

## Author contributions
Conceptualization: F.M.W., S.Z.; Data curation: S.Z.; Formal analysis: S.Z.; Funding acquisition: F.M.W., S.G.-M.; Investigation: S.Z., T.H., A.M., M.E., M.B.; Methodology: S.Z., T.H., M.E., M.B., S.G.-M.; Project administration: F.M.W.; Resources: F.M.W.; Supervision: F.M.W., S.G.-M.; Visualization: S.Z.; Writing – original draft: F.M.W., S.Z.; Writing – review & editing: F.M.W., S.Z., T.H., A.M., M.E., M.B., S.G.-M.

## Funding
This work was funded by grants to F.M.W. from the UK Medical Research Council (MR/PO18823/1), the Wellcome Trust (206439/Z/17/Z; 098503/Z/12/Z) and the Danish National Research Foundation (DNRF135). This work was also supported by an Engineering and Physical Sciences Research Council (EPSRC) Strategic Equipment Grant (EP/M022536/1), Biotechnology and Biological Sciences Research Council (BBSRC) sLoLa (BB/V003518/1), Leverhulme Trust Research Leadership Award (RL 2016-015), Wellcome Trust Investigator Award (212218/Z/18/Z) and Royal Society Wolfson Fellowship (RSWF/R3/183006) to S.G.-M. Open Access funding was provided by the European Molecular Biology Laboratory. Deposited in PMC for immediate release.

## Data and resource availability
All relevant data can be found within the article and its supplementary information. RNA sequencing data are deposited in the Gene Expression Omnibus under accession code GSE303873.

## Peer review history
The peer review history is available online at https://journals.biologists.com/jcs/lookup/doi/10.1242/jcs.264242.reviewer-comments.pdf

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
