## [Peer Review File · Journal of Cell Science]

Cell volume regulates terminal differentiation of cultured human epidermal keratinocytes

Sebastiaan Zijl, Toru Hiratsuka, Atefeh Mobasseri, Mirsana Ebrahimkutty, Mandy Börmel, Sergi Garcia-Manyes and Fiona M. Watt
DOI: 10.1242/jcs.264242

Editor: Andrew Ewald

Review timeline

Submission to Review Commons:	7 December 2024
Submission to Journal of Cell Science:	24 June 2025
Accepted:	30 June 2025

Reviewer 1

Evidence, reproducibility and clarity

Summary: This manuscript investigates the intriguing role of cell volume in regulating keratinocyte differentiation. It provides compelling evidence that reduced cell volume inhibits differentiation, whereas increased cell volume promotes it, across various contexts. The study also examines the effects of substrate topographies on differentiation and cell volume. The findings are interesting and contribute to understanding how cell-niche interactions influence differentiation. However, the manuscript lacks mechanistic insights into how cell volume regulates keratinocyte differentiation. Specifically, the role of transcription factors and signaling pathways in mediating this process remains unexplored.

Major comments:

- It would be valuable to investigate whether changes in cell volume influence the activity of mechanosensitive transcription factors such as YAP/TAZ or SRF/MAL, which are known to regulate differentiation in keratinocytes.
- Keratinocyte differentiation is inhibited on S2 substrates, whether this pause involves regulation of differentiation-associated transcription factors? Including experiments to address this would strengthen the findings.

Significance

Cell volume has emerged as a critical parameter in regulating cell fate decisions, particularly in mesenchymal stem cells, where numerous studies have demonstrated its role in differentiation and self-renewal processes (For example Guo Ming, PNAS, 2017; Bao Min, Nat Commun, 2017; Dudaryeva O Y, Advanced Functional Materials, 2021; Xie Jing, Annual Review of Biophysics, 2024.). However, the influence of cell volume on keratinocyte differentiation had not been explored prior to this study.

Previously, the same research group reported that keratinocyte differentiation is determined by cell spreading area, concluding that FOS and JUNB are SRF target genes essential for terminal differentiation. In their earlier work, serum was shown to stimulate FOS expression, while G-actin levels and MAL activity regulated JUNB expression (Connelly, J., Nat Cell Biol, 2010). However, in the current manuscript, the mechanistic section is missing, it will be interesting to explore whether cell volume regulates keratinocyte differentiation through similar signaling pathways, such as those involving SRF, FOS, and JUNB.

Reviewer 2**Evidence, reproducibility and clarity****Summary:**

This paper explores the role of cell volume as a key regulator in primary cultured human keratinocyte differentiation. Utilizing a unique in vitro platform, the authors show that substrates S1 (associated with larger cell volume and increased differentiation) and S2 (associated with smaller cell volume and reduced differentiation) influence keratinocyte differentiation. The study reveals that changes in cell volume induced by PEG and DI affect differentiation independently of stretch-dependent calcium signaling. While previous research has highlighted physical factors like stretch and stiffness in regulating keratinocyte differentiation states, this study introduces a novel mechanism of cell volume-dependent regulation.

Major comments:

The study's primary conclusion, that keratinocyte differentiation is regulated by cell volume, is well-supported by the experimental data.

Although cell and nuclear volumes were quantified, the nucleus-to-cytoplasm (N/C) ratio was not presented. This ratio might vary between stem and differentiated cells. Observations from Figure 4 suggest that smaller cells might have a higher nuclear ratio, warranting a calculation and discussion of the N/C ratio.

The methodology for excluding cell cycle effects in this analysis needs clarification.

The RNA-seq data compared S1 and S2 conditions, but did not include flat data for comparison.

Since cell volume remains unchanged between flat and S1, micro-pattern- induced gene expression changes could isolate the effects of cell volume from other factors. If previous studies from the author's group have RNA-seq data, re-analysis could be beneficial.

While there were no significant differences in global gene expression between S1 and S2 at 1 and 4 hours, cell volume differences were notable as a cellular phenotype. Although many differentiation-related genes were identified from their RNA-seq data, the primary molecular changes due to cell volume differences remain unclear. Functional classification of genes obtained from RNA-seq data may be useful for differentiating between differentiation-related genes and other possible (direct) regulators.

Is there any cytotoxicity caused by PEG and DI?

It would be informative to examine if aquaporin expression varies among flat, S1, and S2 conditions.

Referees cross-commenting

While all reviewers pointed out the conceptual novelty of this work, they also pointed out that the molecular mechanisms, particularly the relationship with mechanical factors, were unclear. Since it would require a great deal of experimentation to respond to all of the reviewers' points, it is necessary to provide clear answers to which comments can be answered.

Significance

While it is well known that undifferentiated human keratinocytes (epidermal stem cells) are small in size, it remains unknown if this is a cause or consequence of their differentiation status. This study introduces a novel perspective by demonstrating that cell volume may regulate keratinocyte differentiation using a unique in vitro platform developed by the authors, marking a significant advancement in basic cell biology and skin biology.

The molecular mechanisms underlying this regulation are still unknown. Even the summary diagram in Figure 10 lists factors involved in keratinocyte differentiation without clarifying their crosstalk and relationships with known regulators of epidermal differentiation. Future research could address this point by comparing cell volume changes induced by PEG and DI, along with

cells in flat, S1, and S2 conditions, using RNA-seq and other methods.

Questions remain about whether larger cell volume universally leads to differentiation or if this is specific to keratinocytes. If differences exist among cell types, what causes these variations?

Additionally, is this phenomenon exclusive to in vitro conditions, or does it occur in physiological contexts as well? For instance, does cell volume itself influence epidermal differentiation in vivo, and could abnormal volume control or osmotic stress be a primary factor in conditions such as skin inflammation, aging, and cancer?

Audience: basic cell biology, skin biology, stem cell biology, tissue engineering, biophysics

My expertise: epidermal stem cell biology, mouse model, in vitro 3D culture

Reviewer 3

Evidence, reproducibility and clarity

This manuscript by Zijl et al. investigates the role of cell volume regulation in keratinocyte terminal differentiation. The authors use a topography-based approach, comparing two microstructured substrates (S1 and S2) that either promote or suppress differentiation while cells remain spread. They further show that cell volume manipulation (via PEG and DI treatments) can independently regulate differentiation, pointing to cell volume as a critical determinant of fate decisions. The study employs live-cell imaging, RNA sequencing, AFM-based stiffness measurements, and pharmacological inhibitors, making it a methodologically strong contribution to the field of mechanobiology and epidermal stem cell regulation.

Major Comments

1. Lack of a defined molecular mechanism: The study strongly suggests that cell volume influences differentiation, but it remains unclear how volume changes are sensed at the molecular level. The involvement of mechanosensitive transcription factors (e.g., YAP/TAZ, SRF, MRTF-A) should be considered, especially since keratinocyte fate is linked to mechanical signals. The authors mention aquaporin-3 (AQP3) inhibition blocks differentiation, but the exact signaling pathway remains unexplored. Could AQP3 regulate volume-sensitive ion channels (e.g., TRPV4, Piezo1)?

Suggestion: Include a brief discussion or hypothesis on possible molecular sensors linking volume to differentiation.

2. RNA sequencing analysis could be more informative: The RNAseq analysis only identifies differential expression at 12h, missing potential early commitment markers. What about earlier time points (e.g., 6h, 9h)? A transient gene expression shift might precede the 12h differentiation signature.

Suggestion: If additional sequencing is not feasible, the authors could explore existing single-cell datasets on keratinocyte differentiation to identify potential early regulators.

3. Cell stiffness vs. volume relationship is unclear: The AFM data show that cells on both S1 and S2 are softer than on flat substrates, yet only S2 cells exhibit volume reduction and differentiation inhibition. This raises questions about actin cytoskeleton involvement—does actin remodeling contribute to volume-dependent fate regulation?

Suggestion: The authors should clarify whether actin organization differs between S1 and S2 conditions.

Minor Comments: Discussion of topography-specific effects, the authors highlight that differentiation on S1 requires actomyosin activity, yet S2 suppression mechanisms are unclear. Is differentiation inhibition on S2 due to nuclear confinement, actin tension, or substrate geometry effects?

Significance

The findings have broad implications for mechanobiology, skin tissue engineering, and stem cell

differentiation, as they reveal a novel, biophysically-driven mechanism of fate regulation that is independent of traditional biochemical cues. The study identifies cell volume as an independent regulator of differentiation, a new perspective in the field of mechanobiology and epidermal stem cell regulation.

The authors combine live-cell imaging, RNA sequencing, atomic force microscopy (AFM), and pharmacological inhibitors, ensuring a comprehensive, multi-scale analysis. The work has relevance for fundamental biology (cell fate regulation), applied research (skin tissue engineering), and disease modeling (wound healing, hyperproliferative disorders, etc.). The manuscript is well-organized, with strong experimental support for its conclusions.

While the study convincingly links cell volume to differentiation, the precise molecular pathways remain unclear.

The study fills a critical gap in our understanding of how physical properties (cell volume) influence differentiation independently of biochemical signals. It shifts the paradigm by proposing that keratinocyte differentiation is not solely governed by ECM adhesion or substrate stiffness but also by intrinsic volume regulation.

Author response to reviewers' comments

Manuscript number: RC-2024-02827

Corresponding author(s): Fiona Watt

1. General Statement

We feel that our work will be of interest to readers of J. Cell Biol. Because it provides evidence that cell volume regulates differentiation.

Reviewer #1

Evidence, reproducibility and clarity

Summary: This manuscript investigates the intriguing role of cell volume in regulating keratinocyte differentiation. It provides compelling evidence that reduced cell volume inhibits differentiation, whereas increased cell volume promotes it, across various contexts. The study also examines the effects of substrate topographies on differentiation and cell volume. The findings are interesting and contribute to understanding how cell-niche interactions influence differentiation. However, the manuscript lacks mechanistic insights into how cell volume regulates keratinocyte differentiation. Specifically, the role of transcription factors and signaling pathways in mediating this process remains unexplored.

Major comments:

It would be valuable to investigate whether changes in cell volume influence the activity of mechanosensitive transcription factors such as YAP/TAZ or SRF/MAL, which are known to regulate differentiation in keratinocytes.

We have now included data on YAP localization in response to the substrate topographies, PEG and distilled water (Supplemental Figure 5). Nuclear YAP is decreased on both substrate topographies; increases in response to PEG treatment; and is unaffected by DI. We do not, therefore, see a consistent correlation between nuclear YAP and the differentiation of single cells, consistent with our earlier work (Walko, Nat Commun, 2017) but in contrast to the findings of the Piccolo lab (Totaro et al., 2017).

Keratinocyte differentiation is inhibited on S2 substrates, whether this pause involves regulation of differentiation-associated transcription factors? Including experiments to

address this would strengthen the findings.

We agree that this is an interesting question. We previously characterized a subset of JUN (JUNB, JUND), FOS (CFOS, FOSL1) and MAF (MAF, MAFB, MAFF and MAFG) family AP1 factors that are associated with suspension-induced keratinocyte differentiation (see Mishra et al., 2017) and we have now included heatmaps of their expression levels on S1 and S2 substrates (new Figure 3F).

Significance

Cell volume has emerged as a critical parameter in regulating cell fate decisions, particularly in mesenchymal stem cells, where numerous studies have demonstrated its role in differentiation and self-renewal processes (For example Guo Ming, PNAS, 2017; Bao Min, Nat Commun, 2017; Dudaryeva O Y, Advanced Functional Materials, 2021; Xie Jing, Annual Review of Biophysics, 2024.). However, the influence of cell volume on keratinocyte differentiation had not been explored prior to this study.

Previously, the same research group reported that keratinocyte differentiation is determined by cell spreading area, concluding that FOS and JUNB are SRF target genes essential for terminal differentiation. In their earlier work, serum was shown to stimulate FOS expression, while G-actin levels and MAL activity regulated JUNB expression (Connelly, J., Nat Cell Biol, 2010). However, in the current manuscript, the mechanistic section is missing, it will be interesting to explore whether cell volume regulates keratinocyte differentiation through similar signaling pathways, such as those involving SRF, FOS, and JUNB.

We agree with this assessment. While we have included some additional data the full mechanistic analysis will await further experiments.

Reviewer #2

Evidence, reproducibility and clarity

Summary:

This paper explores the role of cell volume as a key regulator in primary cultured human keratinocyte differentiation. Utilizing a unique in vitro platform, the authors show that substrates S1 (associated with larger cell volume and increased differentiation) and S2 (associated with smaller cell volume and reduced differentiation) influence keratinocyte differentiation. The study reveals that changes in cell volume induced by PEG and DI affect differentiation independently of stretch-dependent calcium signaling. While previous research has highlighted physical factors like stretch and stiffness in regulating keratinocyte differentiation states, this study introduces a novel mechanism of cell volume-dependent regulation.

Major comments:

The study's primary conclusion, that keratinocyte differentiation is regulated by cell volume, is well-supported by the experimental data. Although cell and nuclear volumes were quantified, the nucleus-to-cytoplasm (N/C) ratio was not presented. This ratio might vary between stem and differentiated cells. Observations from Figure 4 suggest that smaller cells might have a higher nuclear ratio, warranting a calculation and discussion of the N/C ratio.

This is a good point. We have now included data on nucleus to cytoplasmic ratio after 4h and 12h (new Figure 4F, G).

The methodology for excluding cell cycle effects in this analysis needs clarification.

We added EdU to cells cultured on flat, S1 and S2 substrates and measured the percentage of positive cells after 12h and 24h. There was no difference between the three conditions at 12h; the decrease in S1 cultures at 24h probably correlates with the stimulation of differentiation (new Supplementary Figure 3). We saw no differences in apoptosis, as visualized by cleaved Caspase 3 labelling on the different substrates (new Supplementary Figure 3).

The RNA-seq data compared S1 and S2 conditions, but did not include flat data for comparison. Since cell volume remains unchanged between flat and S1, micro-pattern- induced gene expression changes could isolate the effects of cell volume from other factors. If previous studies from the

author's group have RNA-seq data, re-analysis could be beneficial.

This is a good point. However, the effects of cell volume should be evident at 4h on S2, since volume is unchanged on flat and S1 substrates at 4h.

While there were no significant differences in global gene expression between S1 and S2 at 1 and 4 hours, cell volume differences were notable as a cellular phenotype. Although many differentiation-related genes were identified from their RNA-seq data, the primary molecular changes due to cell volume differences remain unclear.

We have now included heatmaps of expression of aquaporins and genes involved in ion transport and volume regulation, which provide some interesting correlations that we intend to pursue in future (new Figure 9A, B).

Functional classification of genes obtained from RNA-seq data may be useful for differentiating between differentiation-related genes and other possible (direct) regulators.

We have included the GO term data in Supplementary Figure 1.

Is there any cytotoxicity caused by PEG and DI?

We have not directly assessed this. However, we did not see any cleavage products of beta-actin in the Western blots shown in Supplementary Figure 4 (see Kayalar et al., 1996 PNAS 93:2234); we did not see significant Caspase 3 cleavage in cells on S1 and S2 substrates (Supplementary Figure 3); and following washout of PEG300 (Figure 8D) differentiation recovers.

It would be informative to examine if aquaporin expression varies among flat, S1, and S2 conditions.

We agree and have included heatmaps (Figure 9A, B), as described above.

Significance

Significance:

While it is well known that undifferentiated human keratinocytes (epidermal stem cells) are small in size, it remains unknown if this is a cause or consequence of their differentiation status. This study introduces a novel perspective by demonstrating that cell volume may regulate keratinocyte differentiation using a unique in vitro platform developed by the authors, marking a significant advancement in basic cell biology and skin biology.

The molecular mechanisms underlying this regulation are still unknown. Even the summary diagram in Figure 10 lists factors involved in keratinocyte differentiation without clarifying their crosstalk and relationships with known regulators of epidermal differentiation.

Future research could address this point by comparing cell volume changes induced by PEG and DI, along with cells in flat, S1, and S2 conditions, using RNA-seq and other methods.

We agree and have revised the Discussion accordingly.

Questions remain about whether larger cell volume universally leads to differentiation or if this is specific to keratinocytes. If differences exist among cell types, what causes these variations?

Again, this is an excellent point, which we now include in the Discussion. For example, an increase in cell volume induced with distilled water affects differentiation in mesenchymal stem cells (Guo, PNAS, 2017).

Additionally, is this phenomenon exclusive to in vitro conditions, or does it occur in physiological contexts as well?

For instance, does cell volume itself influence epidermal differentiation in vivo, and could abnormal volume control or osmotic stress be a primary factor in conditions such as skin inflammation, aging, and cancer?

There is indirect evidence that this is also important in vivo, as set out in the revised Discussion.

Audience: basic cell biology, skin biology, stem cell biology, tissue engineering, biophysics

My expertise: epidermal stem cell biology, mouse model, in vitro 3D culture

Reviewer #3

(Evidence, reproducibility and clarity (Required)):

This manuscript by Zijl et al. investigates the role of cell volume regulation in keratinocyte terminal differentiation. The authors use a topography-based approach, comparing two microstructured substrates (S1 and S2) that either promote or suppress differentiation while cells remain spread. They further show that cell volume manipulation (via PEG and DI treatments) can independently regulate differentiation, pointing to cell volume as a critical determinant of fate decisions. The study employs live-cell imaging, RNA sequencing, AFM- based stiffness measurements, and pharmacological inhibitors, making it a methodologically strong contribution to the field of mechanobiology and epidermal stem cell regulation.

Major Comments

1. Lack of a defined molecular mechanism: The study strongly suggests that cell volume influences differentiation, but it remains unclear how volume changes are sensed at the molecular level. The involvement of mechanosensitive transcription factors (e.g., YAP/TAZ, SRF, MRTF-A) should be considered, especially since keratinocyte fate is linked to mechanical signals. The authors mention aquaporin-3 (AQP3) inhibition blocks differentiation, but the exact signaling pathway remains unexplored. Could AQP3 regulate volume-sensitive ion channels (e.g., TRPV4, Piezo1)? Suggestion: Include a brief discussion or hypothesis on possible molecular sensors linking volume to differentiation.

We agree that this is a limitation of our study. We have now included data on YAP (Supplementary Figure 5), aquaporins (Figure 9, B), AP1 factors (Figure 3F) and genes involved in regulation of ion transport and cell volume (Figure 9A, B); nevertheless our findings are mainly correlative at this point.

2. RNA sequencing analysis could be more informative: The RNAseq analysis only identifies differential expression at 12h, missing potential early commitment markers. What about earlier time points (e.g., 6h, 9h)? A transient gene expression shift might precede the 12h differentiation signature.

Suggestion: If additional sequencing is not feasible, the authors could explore existing single-cell datasets on keratinocyte differentiation to identify potential early regulators.

We have not included new RNAseq data but have carried out more detailed analysis, as described above.

3. Cell stiffness vs. volume relationship is unclear: The AFM data show that cells on both S1 and S2 are softer than on flat substrates, yet only S2 cells exhibit volume reduction and differentiation inhibition. This raises questions about actin cytoskeleton involvement-does actin remodeling contribute to volume-dependent fate regulation?

Suggestion: The authors should clarify whether actin organization differs between S1 and S2 conditions.

We have previously described actin organization on S1 and S2 substrates (Zijl et al., 2019) and have included the information in the revised Discussion.

Minor Comments: Discussion of topography-specific effects, the authors highlight that differentiation on S1 requires actomyosin activity, yet S2 suppression mechanisms are unclear. Is differentiation inhibition on S2 due to nuclear confinement, actin tension, or substrate geometry effects?

These are excellent suggestions that we are currently exploring. While we have not directly assessed the role of nuclear confinement, we have treated cells on S2 with BAPTA-AM, which blocks intracellular calcium release in response to nuclear deformation (Nava, Cell, 2020). Treatment with BAPTA-AM on S2 did not alter the dynamics of differentiation (Figure 9C), arguing against a role for nuclear deformation on differentiation. Actin tension is likely involved in differentiation, but only on S1 (Zijl, Acta Biomater, 2019), as we have previously observed that treatment with Blebbistatin (an inhibitor of actomyosin contractility) does not reverse differentiation on S2. Substrate geometry and cell spreading could be involved in differentiation (Connelly, NCB, 2010), however, we have found that cells on smaller on S2 as compared to flat and S1. As reduced cell spreading stimulates differentiation, it is unlikely that this reduces differentiation on S2.

Reviewer #3 (Significance (Required)):

The findings have broad implications for mechanobiology, skin tissue engineering, and stem cell differentiation, as they reveal a novel, biophysically-driven mechanism of fate regulation that is independent of traditional biochemical cues. The study identifies cell volume as an independent regulator of differentiation, a new perspective in the field of mechanobiology and epidermal stem cell regulation.

The authors combine live-cell imaging, RNA sequencing, atomic force microscopy (AFM), and pharmacological inhibitors, ensuring a comprehensive, multi-scale analysis. The work has relevance for fundamental biology (cell fate regulation), applied research (skin tissue engineering), and disease modeling (wound healing, hyperproliferative disorders, etc.). The manuscript is well-organized, with strong experimental support for its conclusions. While the study convincingly links cell volume to differentiation, the precise molecular pathways remain unclear.

The study fills a critical gap in our understanding of how physical properties (cell volume) influence differentiation independently of biochemical signals. It shifts the paradigm by proposing that keratinocyte differentiation is not solely governed by ECM adhesion or substrate stiffness but also by intrinsic volume regulation.

Original submission

First decision letter

MS ID#: jcs.264242

MS Title: Cell volume regulates terminal differentiation of cultured human epidermal keratinocytes

Authors: Fiona Mary Watt; Sebastiaan Zijl; Toru Hiratsuka; Atefeh Mobasseri; Mirsana Ebrahimkuty; Mandy Boermel; Sergi Garcia-Manyes

Article Type: Review Commons Transfer

Dear Dr Watt,

I am happy to tell you that your manuscript has been accepted for publication in Journal of Cell Science, pending standard publication integrity checks.

Thank you for sending your manuscript to Journal of Cell Science through Review Commons.